# One Model to Train Them All: A Unified Diffusion Framework for Multi-Context Neural Population Forecasting

## Abstract

Recent research has revealed shared neural patterns among animals performing similar tasks and within individual animals across different tasks. This has led to a growing interest in replacing single-session latent variable models with a unified model that allows us to align recordings across different animals, sessions, and tasks, despite the challenge of distinct neuron identities in each recording. In this work, we present a conditioned diffusion framework to model population dynamics of neural activity across multiple contexts. The quality of the learned dynamics is evaluated through the model's forecasting ability, which predicts multiple timesteps of both neural activity and behavior. Additionally, we introduce a benchmark dataset spanning six electrophysiology datasets, seven tasks, 19 animals, and 261 sessions, providing a standardized framework for multi-task neural population models. Our results demonstrate that the pretrained model can be efficiently adapted to novel, unseen sessions without requiring explicit neuron correspondence. This enables few-shot learning with minimal labeled data, as well as competitive performance in zero-shot learning.

## A  Introduction

Recent advances in neural recording technologies have enabled the simultaneous capture of large populations of neurons, revealing complex spatiotemporal activity. To address this, computational models have been developed to infer latent structures from high-dimensional neural data. Dowling et al. (2023); Pandarinath et al. (2018a); Duncker et al. (2019) Deep generative models, such as variational autoencoders (VAEs) Kingma & Welling (2014) and sequential VAEs, have been widely adopted to extract these latent processes, typically mapping neural or behavioral data to low-dimensional representations Schulz et al. (2024).

Despite the increasing availability of large-scale electrophysiological recordings, current research in neural coding and computation predominantly focuses on single tasks or individual experimental sessions. This current approach misses the opportunity to utilize the structure across individual datasets. Mounting evidence for such possibility, especially in the brain areas related to motor control showed common representation of latent trajectories for stereotypical behavior Gallego et al. (2020); Safaie et al. (2022); Dabagia et al. (2023) and generalizability of dynamical cortical behavior Karpowicz et al. (2022a); Vermani et al. (2024b).

Fitting a single model to a collection of neural data with expected shared features can increase data usage efficiency and enhance our understanding of common neural computation structures that are generalizable. Moreover, this approach can enable efficient scientific progress in new experiments through few-shot learning Ye et al. (2023), where novel experiments could benefit from pre-existing knowledge, reducing the need for extensive new data collection.

Motivated by the success of pre-trained models in machine learning and the growing interest in unified/large scale models for neuroscience Azabou et al. (2023) Ye et al. (2023), we explore the potential of building a foundation model for forecasting neural population data. However, there is a challenge of inherent statistical heterogeneities across datasets such as differences in the number and tuning properties of recorded neurons or variations in recording modalities. Thus, previous approaches uses alignment process to transform new data so that it matches the statistical properties

of the data used to train the model. Aligning neural datasets typically rely on access to the original data used to train the model and/or the existence of paired samples between datasets Pandarinath et al. (2018b); Williams et al. (2021). These paired samples are usually constructed by arbitrarily matching stimulus-conditioned neural activity across datasets, which ignores trial-to-trial variability and is unsuitable for naturalistic tasks. Furthermore, many alignment methods fail to model the temporal structure of neural data, which can lead to suboptimal learning outcomes Wang et al. (2023). Some recent approaches aim to train alignment networks for transferring models across sessions Vermani et al. (2024b), but they remain limited in scope.

In this work, we propose a novel approach that avoids explicit alignment by leveraging the powerful implicit alignment capabilities of conditional diffusion models, a Multi-X Denoising Diffusion Model (Multi-X DDM). Diffusion models have gained significant attention in recent years, originally introduced in the context of image generation condition on a text prompt, has shown remarkable flexibility and effectiveness in wide array of domains, especially in the presence of large and diverse datasets Yang et al. (2023). This work aims to explore the application of these models within the context of neural data analysis.

Diffusion models have demonstrated considerable success in aligning images with textual descriptions through conditioning on contextual information Rombach et al. (2022). We adapt this principle to neural recordings by leveraging the implicit alignment mechanism of diffusion models, allowing for the transfer of pre-trained models across different datasets/tasks without the need for paired samples or explicit alignment procedures. By conditioning our models on relevant context features, we enable zero-shot learning for forecasting neural data, thereby effectively addressing the limitations associated with traditional few-shot learning approaches. Our conditional diffusion model is capable

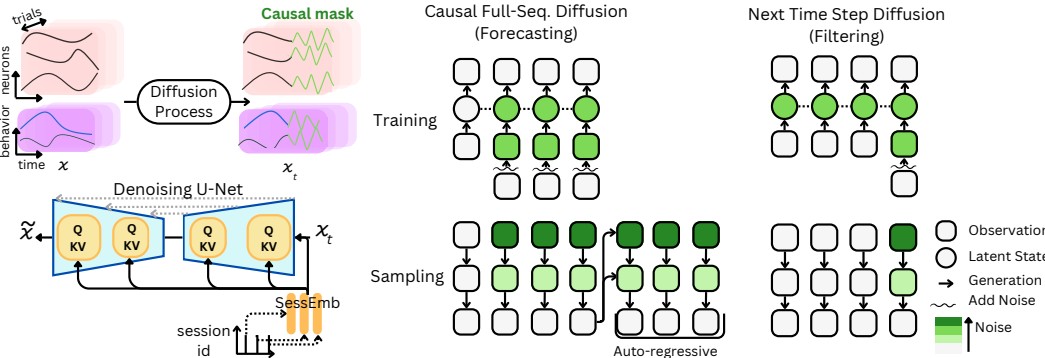

Figure 1: **Multi-X Denoising Diffusion Model overview.** The neural and behavioral data, $x$, are transformed through a diffusion process to $x_t$ with a causal mask enabling causal predictions. The Denoising U-Net, with cross-attention blocks, reconstructs the data conditioned on session embeddings. The model supports both sequence-to-sequence forecasting and filtering.

of forecasting both behavioral and neuronal spike activity from neural data. This integrated approach allows us to capture the complex dynamics between the different modalities of neural information, providing a more comprehensive framework for understanding and predicting neural processes.

Nevertheless, despite these exciting research directions, neuroscience lags behind fields like as computer vision (e.g., ImageNet Deng et al. (2009)) and natural language processing (e.g.,GLUE Wang et al. (2019)) in terms of large-scale benchmark datasets for multi-task,multi-animal, multi-session analysis. This type of analysis involves a unified examination across sessions, subjects, and experimental tasks. Such comprehensive approaches are crucial for achieving a holistic understanding of neuroscience Urai et al. (2022).

**Main Contributions** The key contributions of this work are as follows: (1) We present a novel conditional diffusion model that exhibits improved performance on real-world neural data compared to existing methods. (2) The model is trained on behavioral and neural activity, thereby alleviating the need to treat behavioral prediction as a downstream task. (3) This work introduces the first foundation model for neuroscience that can forecast multiple time steps and perform zero-shot learning. (4) We also introduce a comprehensive benchmark dataset that consolidates six datasets from motor

and sensory cortical areas, encompassing seven tasks across 19 subjects. This dataset is available in Parquet format for faster performance and in NWB format to facilitate integration with other tools commonly used in neuroscience. (5) Furthermore, we provide an API designed to streamline data loading and preprocessing, enhancing accessibility for researchers in the field.

## B    RELATED WORK

**Generative Models for Neural Data Forecasting**  One of the key objectives of computational neuroscience is to build a model capable of reproduce the dynamics of the neural recording. The assessment of this is made by the accurate forecasting capacities of such model.

Classical models, such as Latent Factor Analysis via Dynamical Systems (LFADS) Pandarinath et al. (2018a), use nonlinear recurrent neural networks to infer latent dynamics by modeling neural activity as a dynamical system, generating denoised firing rates from an RNN generator. Similarly, the Structured Variational Autoencoder (SVAE) Johnson et al. (2016); Zhao & Linderman (2023) preserves temporal structure by constraining the prior to a linear dynamical system but faces limitations with nonlinear dynamics. The Deep Kalman Filter (dKF) Krishnan et al. (2016) employs black-box inference networks for joint posterior sampling but encounters difficulties in learning generative dynamics due to issues with gradient propagation. The Deep Variational Bayes Filter (dVBF) Karl et al. (2017) handles state-space graphical models by sampling from the approximate posterior, mitigating the complexity of directly parameterizing dynamics. The eXponential Family Dynamical Systems (XFADS) Dowling et al. (2024) approach extends these ideas to nonlinear Gaussian state-space models, leveraging low-rank approximations and efficient message passing for scalable inference. Although these models are specifically designed to forecast dynamics within a single session, our foundation model is capable of generalizing across multiple sessions and subjects. To evaluate its robustness and effectiveness, we compared its performance against these classical approaches tailored for single-session dynamics.

**Multi-Session Forecasting**  Previous research on multi-session training and alignment has primarily focused on fitting models to data recorded from the same subject, in the same brain region, often with the same chronic implant, and typically assumes a single, stereotyped task structure. Many of these studies do not train on multiple sessions simultaneously, but instead rely on post-training alignment strategies to transfer models trained on one session to other recording days. Pre-trained sequential VAEs (seqVAEs) Vermani et al. (2024a) represent one such approach, introducing an unsupervised alignment mechanism where new neural time series data is aligned with latent dynamics learned from an original dataset. This method transforms the new data to conform to pre-trained latent trajectories without requiring paired samples or the original training data. Another method, ERDiff Wang et al. (2023), enhances this alignment by combining a pre-trained seqVAE with a diffusion model that estimates the density of latent trajectories using spatio-temporal transformers, optimizing the alignment via Sinkhorn divergence. NoMAD Karpowicz et al. (2022b) similarly leverages a pre-trained seqVAE, but fits a multivariate Gaussian to the inferred latent states of the original dataset, training the alignment function to match these states to the new dataset using a KL-divergence loss. In contrast, Cycle-GAN Ma et al. (2023) uses adversarial training to align new sessions with the original data via generative adversarial networks. Finally, Orthogonal Procrustes Schoenemann (1966) aligns datasets by learning a transformation based on paired samples, though it requires access to both the original and new datasets.

**Foundation Models for Neuroscience**  Recently, Azabou et al. (2023) and Ye et al. (2023) showed that training not only across subjects but on data from different tasks and laboratories, is possible. However, these models were specifically designed for decoding in brain-machine interface (BMI) applications and lack demonstrated forecasting capabilities, or such capabilities have not been thoroughly explored. It is important to note that NDT2 Ye et al. (2023) has established the potential for zero-shot learning within this context. A comprehensive comparison with our approach is provided in Appendix A.

**Tools for Foundation Models: Benchmark Datasets**  Recent advancements in neuroscience have led to the emergence of large datasets and a focus on data sharing through initiatives like Neurodata Without Borders (NWB) Teeters et al. (2015); Rübel et al. (2022), which enhance reproducibility. It is a growing ecosystem around NWB format hosted on Dandi archive[1]. Which makes the data

---

[1] https://www.dandiarchive.org/

preprocessing easier with tools like Pynapple Viejo et al. (2023) or easy to visualize with Neurosift[2]. However, integrating diverse datasets remains challenging due to varying file structures, and also data formats Pierré et al. (2024). The Neural Latent Benchmark (NLB) Pei et al. (2021) provides a standardized framework for evaluating latent variable models on neural data, focusing on single-session models across four curated datasets. In contrast, our dataset expands on this by addressing multi-session, multi-task, and multi-subject neural data, offering a more comprehensive perspective on neural population dynamics for foundation model development.

## C METHODS

This study utilizes a diffusion model conditioned on *session identifiers*. The diffusion process transforms each training sample into a Gaussian distribution through an iterative noise corruption process, as described in Ho et al. (2020); Rombach et al. (2022). Subsequently, a deep neural network is trained to invert this transformation, enabling the generation of new samples starting from Gaussian noise inputs.

### C.1 CONDITIONAL DENOISING DIFFUSION MODELS

Let $\mathcal{D}$ denote a dataset comprising $M$ trials of neural and behavioral recordings, represented as $\mathcal{D} = \{x_i \mid i = 1, 2, \ldots, M\}$, where $x_i$ corresponds to the data for the $i$-th trial. Each trial includes the smoothed spiking activity of $n$ neurons over $t_1$ *neural activity* time steps, represented as $s \in \mathbb{R}^{n \times t_1}$, along with $m$ behavioral covariates, represented as $b \in \mathbb{R}^{m \times t_1}$. Thus, a sample $x_i$ is given by $x_i = \{s_i, b_i\}$. To jointly generate neural activity and behavior from a shared latent representation, as introduced in Schulz et al. (2024), we eliminate the assumption that behavior is linearly decoded from spiking activity, enabling improved modeling capabilities.

**Forward Process:** The forward noising process $q$ is defined as a Markovian Gaussian noise addition, with noise applied at each *diffusion* time step $t = 1, 2, \ldots, T$ according to a variance schedule $\beta_t$:

$$q(\mathbf{x}_t|\mathbf{x}_{t-1}) = \mathcal{N}(\mathbf{x}_t; \sqrt{1 - \beta_t}\,\mathbf{x}_{t-1}, \beta_t\mathbf{I}), \tag{1}$$

where $\mathcal{N}(\cdot)$ denotes a Gaussian distribution, and $\mathbf{I}$ is the identity matrix. The variance schedule $\beta_t$ is chosen using a cosine scheduler as proposed in Nichol & Dhariwal (2021), which optimizes training efficiency. The joint distribution over all time steps is given by $q(\mathbf{x}_1, \ldots, \mathbf{x}_T|\mathbf{x}_0) = \prod_{t=1}^{T} q(\mathbf{x}_t|\mathbf{x}_{t-1})$. For any timestep $t$, the forward process $q(\mathbf{x}_t|\mathbf{x}_0)$ can be expressed in closed form as $q(\mathbf{x}_t|\mathbf{x}_0) = \mathcal{N}(\mathbf{x}_t; \sqrt{\bar{\alpha}_t}\,\mathbf{x}_0, (1 - \bar{\alpha}_t)\mathbf{I})$, where $\alpha_t = 1 - \beta_t$ and $\bar{\alpha}_t = \prod_{i=1}^{t} \alpha_i$. Consequently, $\mathbf{x}_t$ can be sampled directly as $\mathbf{x}_t = \sqrt{\bar{\alpha}_t}\,\mathbf{x}_0 + \sqrt{1 - \bar{\alpha}_t}\,\boldsymbol{\epsilon}$, where $\boldsymbol{\epsilon} \sim \mathcal{N}(0, \mathbf{I})$. For sufficiently large $T$, it holds that $\mathbf{x}_T \sim \mathcal{N}(0, \mathbf{I})$, corresponding to pure Gaussian noise.

**Training Objective:** Let $\theta$ denote the parameters of the neural network used to approximate the reverse diffusion process. The loss function is defined as:

$$\mathcal{L}_\theta = (1-\lambda)\left\|\boldsymbol{\epsilon}_s - \boldsymbol{\epsilon}_{\theta,s}\left(\sqrt{\bar{\alpha}_t}\mathbf{s}_0 + \sqrt{1-\bar{\alpha}_t}\boldsymbol{\epsilon}_s, y, t\right)\right\|^2 + \lambda\left\|\boldsymbol{\epsilon}_b - \boldsymbol{\epsilon}_{\theta,b}\left(\sqrt{\bar{\alpha}_t}\mathbf{b}_0 + \sqrt{1-\bar{\alpha}_t}\boldsymbol{\epsilon}_b, y, t\right)\right\|^2, \tag{2}$$

where $\boldsymbol{\epsilon}_s \in \mathbb{R}^{n \times T}$ and $\boldsymbol{\epsilon}_b \in \mathbb{R}^{m \times T}$ denote the noise terms for spiking activity and behavioral covariates, respectively. The conditioning variable $y$ encodes the session embedding, while $\lambda \in [0, 1]$ balances the terms.

**Sampling Procedure:** The sampling process begins with $\mathbf{x}_T \sim \mathcal{N}(0, \mathbf{I})$, and iteratively refines this through the reverse process:

$$\mathbf{x}_{t-1} = \sqrt{\bar{\alpha}_{t-1}}\left(\frac{\mathbf{x}_t - \sqrt{1 - \bar{\alpha}_t}\,\boldsymbol{\epsilon}_\theta^{(t)}(\mathbf{x}_t, y, t)}{\sqrt{\bar{\alpha}_t}}\right) + \sqrt{1 - \bar{\alpha}_{t-1} - \sigma_t^2}\,\boldsymbol{\epsilon}_\theta^{(t)}(\mathbf{x}_t, y, t) + \sigma_t\mathbf{z}, \tag{3}$$

where $\sigma_t$ is a constant specific to $t$, and $\mathbf{z} \sim \mathcal{N}(0, \mathbf{I})$. For efficient sampling, we employ Denoising Diffusion Implicit Models (DDIM) as described in Song et al. (2022), which offer superior performance compared to traditional samplers.

---

[2]https://neurosift.app/?p=/dandi

## C.2 Network Architecture

The proposed network $\epsilon_\theta(\mathbf{x}_t, session\ id, t)$ processes noisy inputs $\mathbf{x}_t = \{\mathbf{s}_t, \mathbf{b}_t\}$, embedding *diffusion* temporal information through $\phi_t(t)$ and incorporating session-specific context.

**Temporal Embedding:** $t$ is embedded into a high-dimensional vector $\phi_t(t)$ using sinusoidal positional encodings followed by a learnable projection via an MLP. This embedding is projected to match the feature dimensions and is applied to reweight feature maps after convolutional processing.

**Convolutional Layers:** The network consists of $L$ stacked convolutional blocks, where intermediate feature maps $\mathbf{h}^{l-1}$ are processed as: $\mathbf{h}^l = \text{ReLU}\left(\text{BatchNorm}\left(W_{\text{conv}}^l * \mathbf{h}^{l-1} + b_{\text{conv}}^l\right)\right)$, where $W_{\text{conv}}^l$ is the convolution kernel, $b_{\text{conv}}^l$ is the bias term, and $*$ denotes the convolution operation.

**Residual Connections:** Residual connections enhance feature reuse and gradient flow, defined as: $\mathbf{h}^{\text{res}} = \mathbf{h}^{\text{in}} + g(\mathbf{h}^{\text{in}}, \phi_t(t))$, where $g(\cdot)$ consists of group normalization, nonlinear activation, and $L$ convolution operations.

**Self-Attention:** To capture global dependencies across all neurons, self-attention is applied after the ResNet block: $\mathbf{h}^{\text{att}} \leftarrow \text{Attention}(Q, K, V) = \text{softmax}\left(\frac{QK^\top}{\sqrt{d_k}}\right)V$, where $Q, K, V$ are linear projections of $\mathbf{h}^{\text{res}}$, and $d_k$ is the scaling factor. This enables neurons to attend to one another effectively.

**Session Embedding and Guidance:** Variability in experimental setups, as well as the fact that each session corresponds to a distinct population of neurons, is accounted for by assigning a unique $d$-dimensional embedding to each session: $y_i = \text{SessionEmbed}(session\ id\ of\ the\ i\text{-}th\ trial)$, where $y_i \in \mathbb{R}^d$ is integrated into the network via cross-attention. The cross-attention mechanism is defined as: $\text{Attention}(Q, K, V) = \text{softmax}\left(\frac{QK^\top}{\sqrt{d}}\right)V$, with $Q = W_Q^{(i)} \cdot \mathbf{h}^{\text{att}}, \quad K = W_K^{(i)} \cdot y^{(i)}, \quad V = W_V^{(i)} \cdot y^{(i)}$. Here, $W_Q^{(i)}, W_K^{(i)}, W_V^{(i)}$ are learnable projection matrices.

**Multi-Resolution Processing:** After each hierarchical block (comprising $L$ convolution layers, ResNet, self-attention and cross-attention), the spatial and temporal dimensions are reduced using stride-2 convolutions. The network performs multi-resolution processing by hierarchically downsampling across $H$ blocks and then upsampling with stride-2 transposed convolutions.

**Intuition.** Each hierarchical block captures information at a specific resolution, enabling the model to effectively represent both long- and short-term dependencies. Compared to dilated convolutions used in WaveNet, stride-2 convolutions are computationally efficient and avoid exponential growth in receptive field sizes. Residual connections ensure critical information is preserved during downsampling, while self-attention mechanisms optimize dimensionality reduction without losing key relationships. Cross-attention mechanisms guide the generation process, adapting it to each session.

## C.3 Causal Forecasting Masking

Building on the foundation of the multi-task masking approaches proposed in Zhang et al. (2023), we employed a causal mask where we only corrupt the input after a specific time bin to enable causal prediction. This means that the model can only utilize information up to the current time step to forecast the future, which aligns with the requirements of real-world applications.

## C.4 Strategies for Session Transfer

Inspired by classifier-free guidance, our model incorporates both unconditional and conditional components. For each session, 20% of the trials are randomly designated as unconditional. This involves assigning a session ID that is shared across all sessions. During sampling, we can apply the strategy introduced by Ho & Salimans (2022), expressed as $\epsilon_\theta(\mathbf{x}_t, y, t) = \epsilon_\theta(\mathbf{x}_t, t) + g\left(\epsilon_\theta(\mathbf{x}_t, y, t) - \epsilon_\theta(\mathbf{x}_t, t)\right)$, where $\epsilon_\theta(\mathbf{x}_t, y, t)$ is the noise prediction from the model conditioned on $y$, $\epsilon_\theta(\mathbf{x}_t, t)$ is the noise prediction from the unconditioned model, $g$ is the guidance scale that determines the strength of the conditioning.

To generalize to new sessions, we can use the unconditional model with three strategies:

**- Zero-shot learning:** The unconditional model, i.e. conditioned on the shared session ID, predicts held-out sessions without requiring test data alignment, unlike previous methods. However, this may lead to poorer performance since the model has not been adapted to the new session data.

**- Fine-tuning:** The pre-trained model's weights are fine-tuned using a few samples from the new sessions to adapt to their data distributions.

**- Session identification:** Following the *unit identification* strategy of Azabou et al. (2023), we adapt session embeddings using gradient descent, initializing new embeddings with the unconditional session embedding to avoid poor performance from random initialization. This approach maps new sessions into the embedding space by freezing the model's weights and adding rows to the session embedding lookup table, allowing rapid transfer to new sessions. In our model, this strategy closely parallels the fine-tuning of the alignment networks in Vermani et al. (2024b).

### C.5 FORECASTING AND FILTERING

Full-sequence diffusion, widely used in video generation, trains models to denoise entire sequences in a seq2seq manner, allowing for globally optimal predictions within a specified length. This approach supports flexible-length sequence generation by autoregressively using the output without the need for beam search, as required by next-token prediction models like transformers. To enable direct comparison with such models, we trained our approach to predict the next time step (filtering) from a ground-truth sequence using teacher forcing.

## D EXPERIMENTS AND RESULTS

In this section, we showcase the potential of our approach for large-scale training and explore the advantages of scaling to multi-contexts in neural population forecasting.

### D.1 BENCHMARK DATASET AND EXPERIMENT SETUP

One of the key advantages of our approach is its ability to scale to handle large amounts of neural data, including sessions from different numbers of neurons, across different tasks and recording setups, and from different animals. Thus we set out to build a diverse dataset large enough to test our approach. We curated a multi-lab dataset with electrophysiological recordings from motor cortical regions, where neural population activity has been extensively studied, and deep learning tools and benchmarks have recently been established.

A core innovation of our dataset is the new schema that allows to efficiently managing data from different sessions at the same data frame. The dataset was curated from various sources and data

| Dataset ID | Animal | Session | Trial ID | Neurons | | | | | | vx | vy | ... | Target Onset | Go Cue | ... |
|---|---|---|---|---|---|---|---|---|---|---|---|---|---|---|---|
| 1 | 1 | 1 | 1 | 0 | 0 | 1 | 1 | 0 | NaN | 0 | 0 | | | | |
| | | | | 0 | 1 | 0 | 0 | 1 | NaN | 0.001 | 0.001 | | True | | |
| | | | | 1 | 1 | 0 | 0 | 0 | NaN | 2.001 | 1.002 | | | True | |
| | | | | 1 | 1 | 1 | 0 | 0 | NaN | 2.500 | 1.500 | | | | |
| | | | | 1 | 0 | 0 | 1 | 1 | NaN | 3.000 | 2.000 | | | | |
| | | | | 0 | 0 | 1 | 1 | 1 | NaN | 4.000 | 2.500 | | | | |
| | | | 2 | 1 | 0 | 1 | 0 | 0 | 1 | 0 | 0 | | True | True | |
| | | | | 0 | 0 | 0 | 0 | 1 | 0 | 1.010 | 0.500 | | | | |
| | | | | 0 | 1 | 0 | 0 | 0 | 0 | 2.003 | 0.800 | | | | |
| | | | | 1 | 1 | 0 | 1 | 0 | 1 | 4.002 | 1.600 | | | | |
| | | ... | 1 | | | | | | | | | | | | |
| | | | ... | | | | | | | | | | | | |
| | ... | | | | | | | | | | | | | | |
| 2 | 1 | 1 | 1 | | | | | | | | | | | | |
| ... | | | | | | | | | | | | | | | |

Figure 2: Proposed benchmark dataset schema, comprising indexes, neurophysiological data, behavior covariates, and events time indications. Each row represents a single time step.

formats (MATLAB, NWB) into a unified format. We offer two formats for this dataset: Parquet, which will be made available on Kaggle, and NWB, which will be accessible via the Dandi Archive. The Parquet format is optimized for deep learning applications, enabling fast performance, while the NWB format ensures interoperability with a wide range of tools within the NWB ecosystem commonly used in neuroscience research. The dataset comprises six sub-datasets, comprising 261 sessions from 19 monkeys engaged in seven distinct tasks. These sub-datasets collectively enables the evaluation of models which are capable of align between these different scenarios. A detailed description of the full dataset suite is provided in Appendix C.

**Experiment setup.** In all experiments, the datasets were binned at 20 ms, except for Dataset 3, which has a default bin size of 30 ms that was not adjusted. The data were aligned to movement

onset, and the binned spikes were subsequently smoothed using a 50 ms Gaussian kernel. We use the ADAM optimizer, and employ a cosine decay of the learning rate at the end of training. We use 1-GPU setup with 48GB. All details can be found in Appendix B.

### D.2 TESTING THE MODEL ON SINGLE SESSIONS

We evaluated our model on neural recordings collected during a maze task, assessing its ability to forecast both neural spiking and behavioral covariates. The Maze datasets consist of recordings from the primary motor (M1) and dorsal premotor cortices while a monkey performed reaches with an instructed delay to visually presented targets, navigating a virtual maze Churchland et al. (2010). For baseline comparisons, we used the nlb-maze dataset Churchland & Kaufman (2022) for forecasting and filtering modes (Figure 3).

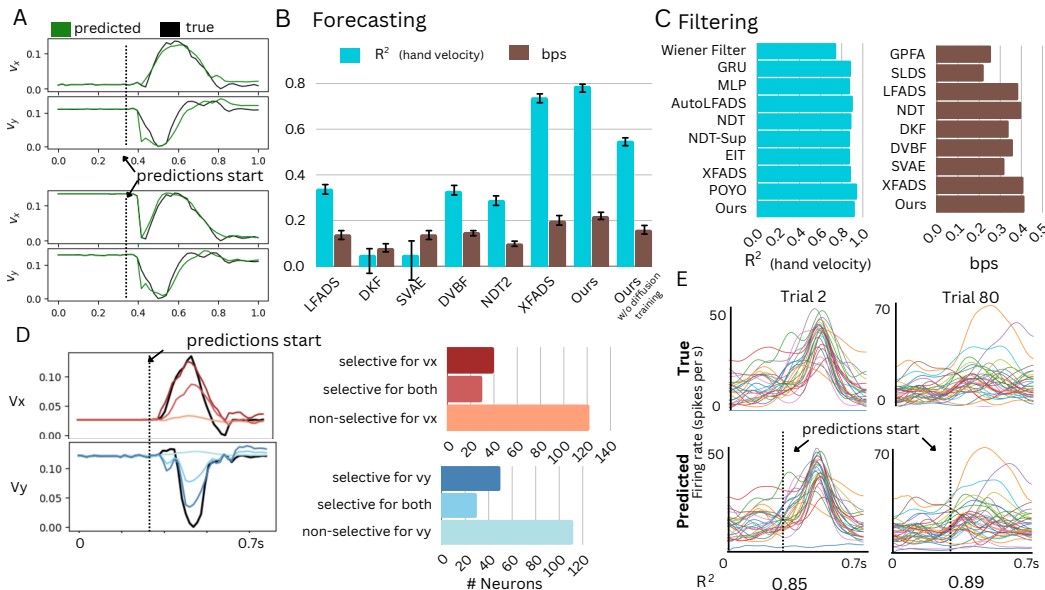

Figure 3: **Validation of the single session model for forecasting and filtering.** (A) Example of $x$ and $y$ hand velocities for 1s.(B) $R^2$ comparison of predicted hand velocity across methods, using a 250 ms context window and a 500 ms forecasting window, with our model using only spiking activity data + linear decoder. And the bits-per-spike (bps) using inferred spike-train rate. (C) Behavior decoding and neural activity prediction results in filtering mode. (D) Predicted $x$ *(top)* and $y$ *(bottom)* velocities. The black trace denotes the ground truth, while colored traces represent the model's predictions. The bar charts on the right display neuron selectivity: neurons selective for $vx$ (red), $vy$ (blue), both (lighter shades), or non-selective (orange/cyan) for each velocity component. (E) Example of neural activity forecasting in two trials for each neuron.

Our model consistently outperformed existing neural forecasting methods such as LFADS, DKF, SVAE, DVBF and NDT2, with performance comparable to XFADS, Fig. 3B. The poor performance of NDT2 with a causal mask was expected, since recent work on transformer architectures has shown that learning global dependencies in temporal tokens leads to poor prediction results on multivariate time series datasets Liu et al. (2024). Thus, this model is not competitive in terms of forecasting and, in fact, that was not its original purpose, nor does POYO's architecture allow for forecasting, so our model is the only unified model proven to be able to so competitively with state-of-the-art neural data forecasting models. To test whether the improved performance is due to our architecture or the training strategy, we trained the model without it (see Appendix B.2). We found that the architecture itself is better and that training with diffusion improves it. We also show that our model is competitive in filtering, Fig. 3C.

The advantage of joint prediction lies in its ability to analyze neuron selectivity for behavior covariates and the timing of this selectivity, both through occlusion sensitivity, Fig. 3D3. We observed selectivity primarily 200-250 ms before movement onset, consistent with our Ablation Studies.

## D.3 TRANSFERRING TO NEW SESSIONS

We validate our framework on held-out sessions from a different animal. The datasets used for these experiments consist of neural recordings obtained from M1 in two monkeys during a center-out (CO) reaching task. One dataset was used for training or pretraining, while the other was used for testing, with both monkeys performing the same task in the same experimental setup. The task required the animals to repeatedly reach toward one of eight predefined targets upon receiving a cue and then return to the center. Each trial began with the subject positioning its hand at the center of the workspace, followed by a random delay before one of the eight peripheral targets was displayed.

We compare the proposed approach against state-of-the-art explicit alignment methods (Table 1). For the single-session experiments, we trained the models using session 4 from Monkey 1 in dataset 5 of our benchmark. To evaluate alignment performance on held-out subjects and sessions, we used session 2 from Monkey 1 and sessions 1 and 2 from Monkey 2 as the test set. For the "all

Table 1: Forecasting performance comparison between methods of 100 ms. The values indicate the median and standard error over the observations from new sessions.

| | Method | $R^2$ |
|---|---|---|
| **One Session** | ERDiff | $-0.23 \pm 0.55$ |
| | NoMAD | $0.15 \pm 0.10$ |
| | Cycle-GAN | $-0.81 \pm 0.12$ |
| | Procrustes | $0.07 \pm 0.14$ |
| | SeqVAE | $\mathbf{0.39} \pm 0.07$ |
| | Multi-x DDM + 0-shot | $\mathbf{0.38} \pm 0.05$ |
| | Multi-x DDM + Session ID | $\underline{0.35} \pm 0.07$ |
| | Multi-x DDM + Fine-tune | $\mathbf{0.42} \pm 0.03$ |
| **All Sessions** | Multi-x DDM + 0-shot | $0.46 \pm 0.03$ |
| | Multi-x DDM + Session ID | $\underline{0.51} \pm 0.04$ |
| | Multi-x DDM + Fine-tune | $\mathbf{0.55} \pm 0.05$ |

sessions" experiments, we trained our Multi-x DDM with all sessions of subject 1 except session 2. In this scenario, there is no correspondence between units in the training and testing conditions. The prediction window for these experiments was set to 100 ms. After pretraining, our model can be tested on new sessions with unknown neurons using either (i) a session identification approach (ii) zero-shot learning or (iii) full fine-tuning, as described in Section C.4.

The results indicate that only SeqVAE and our method achieve good forecasting performance, while other approaches struggle. By training with additional sessions, we can further enhance accuracy, surpassing the performance of single-session models. This underscores the robustness of our model and its flexibility to incorporate new data, whether through a simple input mapping (session identification) or without seeing any data of the test set.

Interestingly, when trained with only one session, zero-shot learning outperforms session identification; however, this trend reverses when more data is available. Our transfer strategies aim to learn shared features while capturing session-specific styles through embeddings. Training on one small session, however, is insufficient to build a noise-robust model, and few-shot examples from another animal fail to separate shared and session-specific features, pushing the model further from the data manifold and amplifying noise in the learned representation. As noted by Vermani et al. (2024b), the present training session itself is highly informative about the testing sessions, which we hypothesize explains why zero-shot learning performs better in this scenario. With sufficient data, the model learns robust shared features, and few-shot examples capture session-specific styles, improving performance. This emphasizes the importance of adequate data for robust representations. While Azabou et al. (2023) showed that large-scale pretraining enables session identification to adapt effectively, our results highlight that without robust pretraining, this does not hold true.

In Fig. 4, we trained a multi-session version of our model using data from all sessions of the same monkey, showing improved neural spiking prediction over the single-session model. Session embeddings clustered by behavioral strategy, despite no explicit training on this. We hypothesize that the clustering makes sense, as faster responses, linked to automatic movements via basal ganglia pathways, differ from slower behaviors, suggesting distinct neural patterns in M1. Notably, the shared ID (0) is positioned between the two clusters.

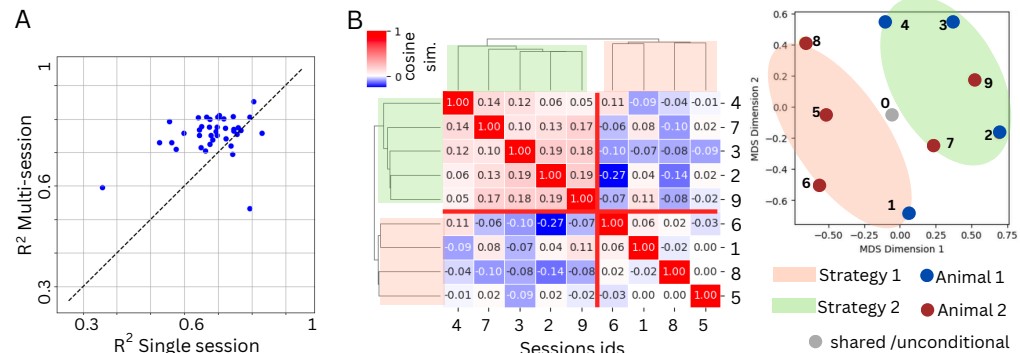

Figure 4: (A) Comparison of neural spiking prediction performance between the single-session and multi-session models. (B) MDS projection of the learned session embeddings for the benchmark dataset 6 and corresponding cosine similarity matrix. Two strategies were identified: **1:** fast reaction time with higher failed trials, and **2:** slower reaction time with higher task success.

### D.4 A MULTI-SESSION, MULTI-ANIMAL, MULTI-TASK AND MULTI-LAB MODEL

In light with the good transfer results of Multi-X DDM for forecasting held-out sessions, we explore the development of a model that spans a even more broader range of recording setups. Thus, to build the multi-task model, we trained it with all sessions from the benchmark dataset, reserving the same held-out sessions as in Section D.3 for testing: one session from one of the animals in the training set (Cross Session analysis) and two sessions from a different animal (Cross Subject analysis), see Figure 5. Despite the considerable variability among the different datasets that constitute

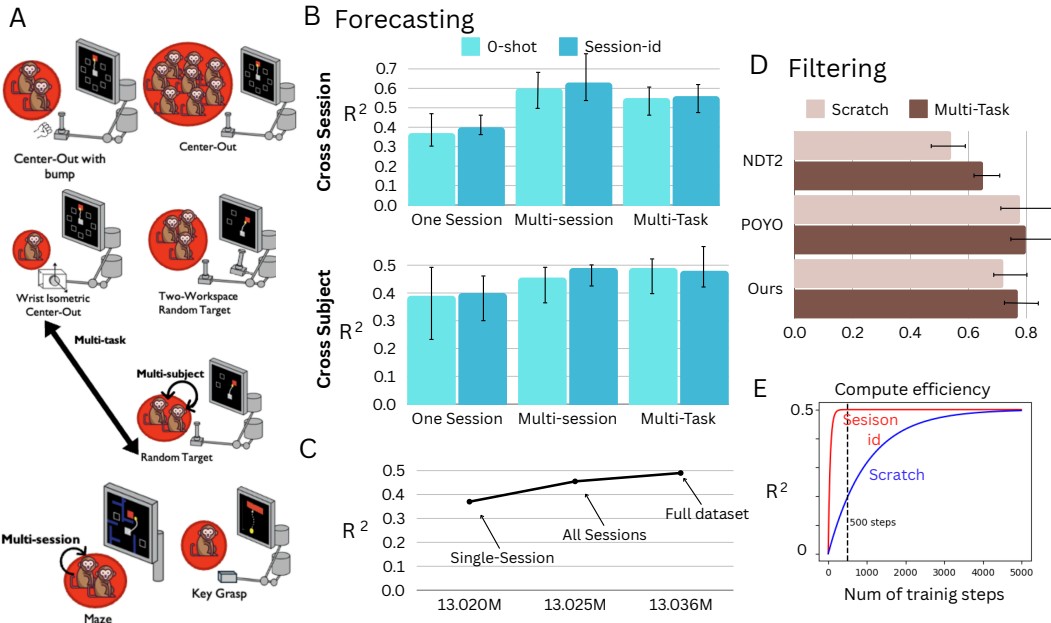

Figure 5: **Scaling up to more tasks and diverse neural and behavioral recording conditions.** (A) Overview of our benchmark dataset used to train the Multi-task model. It combines recordings from multiple monkeys performing various tasks from various labs. (B) Forecasting behavior performance on 500 ms window. Tested one a held-out monkey for transfer testing. Trained with *one session*, all sessions of the subject, *multi session*, and with the full dataset, *multi-task*. (C) Cross-subject Scaling Analysis. (D) Comparison between foundation models for filtering on dataset 1 trained from scratch and with the full dataset. (E) Compute efficiency for training and session identification approaches across subjects.

our benchmark, multi-X DDM provides consistent improvements over the single-session version, as confirmed by our scaling analysis. When comparing multi-session with multi-task, we see that the multi-task generalizes better across different subjects, while the multi-session one excels on the dataset sourced from the same animal. It has been confirmed that in filtering mode, our model is competitive with previous foundation models. These models are effective tools for brain-machine interfaces due to their strong decoding performance and adaptability to real-time settings. However, our model has a different goal: to forecast neural activity and behavior, offering interpretable insights into an animal's strategies and the role of specific neurons in behavior.

## E ABLATIONS

In order to evaluate the effects of training with both spiking activity and behavior data, as well as the impact of data preprocessing and forecasting capabilities, we run a number of ablation experiments, as shown in Fig. 6.

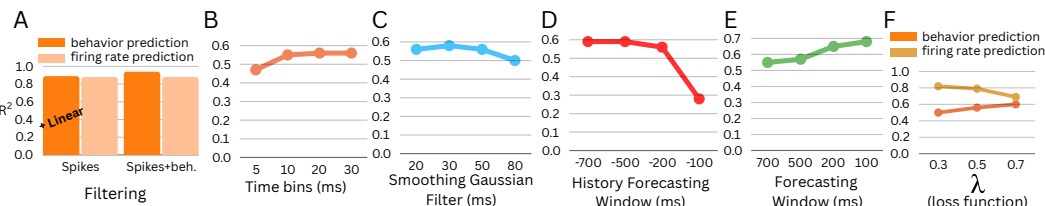

Figure 6: **Ablation studies.** (A) Training with spikes vs. spikes+behavior. (B-E) Effects of time bin size, smoothing filter, history window, and forecasting window on $R^2$ for behavior forecasting. (F) Balance between behavior and neural activity terms in the loss function ($\lambda$).

Panel A of Fig. 6 provides a direct comparison between our joint model and the two-step model, where behavior is predicted using ridge regression based on the next-step spiking activity prediction (filtering). The results show that the joint model outperforms the two-step model in behavior prediction, while maintaining the quality of spiking activity prediction. Thus the joint model effectively learns the relationship between spiking and behavior in an integrated manner, optimizing for both simultaneously. Unlike the two-step model, which separates the tasks of spikes and behavior prediction, the joint model captures the interactions between neural activity and behavior more effectively, leading to improved overall performance. Importantly, the joint model does not sacrifice the accuracy of spike predictions, suggesting that it can handle multiple objectives without degradation.

Additionally, the model demonstrated robustness to different preprocessing methods, including time binning and Gaussian smoothing filters. The analysis of the history forecasting window is consistent with our occlusion analysis, which suggests that the majority of the behavioral signals (hand velocity) occur around 200-250 ms prior to movement onset. Given that we tested forecasting after movement onset in a stereotypical task, we did not observe significant performance degradation as the forecast horizon was extended during the motion phase.

Finally, the lambda term in the loss function significantly influences model performance. In our benchmarks, we found that a value of 0.5 worked well for both behavior and spiking activity. However, this may vary depending on the specific brain area or task being studied.

## F CONCLUSIONS

We present a novel framework for training diffusion models on large, multi-context neural activity datasets, capable of zero-shot learning and providing interpretable embeddings, supported by a benchmark dataset and API for community use. Our findings show that larger training datasets improve the model's ability to predict behavioral outcomes and neural spiking activity, capturing brain dynamics more precisely. Robust zero-shot performance on held-out sessions highlights the model's capacity to learn shared dynamics across sessions and animals. Future work will focus on leveraging explainable machine learning to analyze neural mechanisms, explore population dynamics, and investigate inter-session variability through the session embedding space.

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
