# APPENDIX

## Overview:

- Appendix A contains additional comparison with our framework and previous work

- Appendix B contains additional implementation details for our experiments

- Appendix C contains additional details and plots for benchmark dataset

- Appendix D contains the API functions

- Appendix E contains the dataset documentation

## A  EXTENDED RELATED WORK

There is growing interest in moving beyond filtering models, which only decode the next time step based on a sequence of neural activity, to models capable of forecasting. The key motivation for this shift is that the ability to forecast is an indication that the model is learning the underlying dynamics of the system. While filtering models can perform well, the dynamics they learn may be poor approximations of the true neural dynamics, as observed in various models presented in Figure 3B and Table 1 of the main manuscript. To the best of our knowledge, we are the first to build a unified framework that explicitly demonstrates and is specifically designed for forecasting capabilities.

Table 1: **A non-exhaustive summary of our model capabilities.** The question mark represents theoretically possible but unverified.

| Method | Filtering | Forecasting | Few-shot | 0-shot | Multi-session | Scalable |
|--------|-----------|-------------|----------|--------|---------------|----------|
| XFADS  | ✓ | ✓ | ✗ | ✗ | ✗ | ✗ |
| SeqVAE | ✓ | ✓ | ✓ | ✗ | ✓ | ✗ |
| POYO   | ✓ | ✗ | ✓ | ✗ | ✓ | ✓ |
| NDT2   | ✓ | ✓ | ✓ | ✓ | ✓ | ✓ |
| Ours   | ✓ | ✓ | ✓ | ✓ | ✓ | ✓ |

Other unified models, such as POYO and NDT2, excel in filtering mode, but our model remains highly competitive. These models offer excellent solutions for brain-machine interfaces due to their adaptability and fast inference, making them suitable for real-time applications. In real-time control scenarios, our model can also be applied if one wishes to relax the control over every single step. For instance, using model predictive control, we can control every 5 time steps, and in this case, forecasting capabilities become essential. Our model's ability to predict the optimal next multi-step sequence (e.g., 5 steps) at once significantly speeds up real-time control processes.

Additionally, we extended POYO's session identification strategy to enable 0-shot learning. However, this strategy cannot be directly applied to POYO, as it performs unit alignment in addition to session alignment. Thus, even with shared sessions, POYO still requires registering the new neural population, meaning it always needs a few shots of data from the new session or neural population. Furthermore, POYO is not capable of forecasting, as it only predicts behavior and cannot autoregressively forecast. Regarding NDT2, they reported negative results on held-out neural reconstruction, and while forecasting is theoretically possible, their model has not been demonstrated to do so effectively. Thus, while other methods are highly effective for decoding, our model stands out in forecasting applications.

From a scientific perspective, we demonstrate that our model can extract non-trivial information through the informative session embedding space and the neural activity-behavior selectivity analysis.

# B  ADDITIONAL MODEL DETAILS

## B.1  TRAINING DETAILS

The model is trained using the ADAM optimizer with weight decay. And with mixed precision optimizing hardware efficiency. The learning rate is held constant, $1 \times 10^{-4}$, then decayed towards the end of training (last 25% of epochs), using a cosine decay schedule. Single-session models are trained with a batch size of 100 while large models are trained with a total batch size of 200. Note that we didn't see any benefits in increasing the batch size when training single-session models.

## B.2  WITHOUT DIFFUSION TRAINING

To evaluate our model without diffusion training, we modified the architecture by removing the time embedding and trained it on the entire dataset using a causal mask input. After movement onset, the input consisted of Gaussian noise, and the model was required to reconstruct the original signal. This setup effectively exposed the model exclusively to the most challenging denoising scenarios. As illustrated in the figure below, training without diffusion required substantially more training iterations and consistently yielded lower prediction accuracy. These findings align with previous studies emphasizing the importance of intermediate noise steps in diffusion training for enhancing model prediction quality Nichol & Dhariwal (2021).

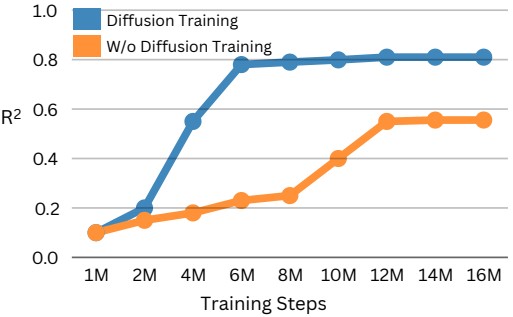

Figure 1: Performance comparison of the model with and without diffusion training.

## B.3  HYPERPARAMETERS

We provide an overview of the hyperparameters of all trained Multi-X DDM models in Tab.

Table 2: Hyperparameters for the conditional Multi-X DDM.

|  | Single Session | Multi-X sessions |
| --- | --- | --- |
| $H$ hierarchical layers | 4 | 4 |
| $L$ convolution layers | 2 | 2 |
| Diffusion steps | 1000 | 1000 |
| Noise Schedule | cosine | cosine |
| Channels | 1 | 1 |
| History window | 250 ms | 500 ms |
| Forecasting window | 500 ms | 500 ms |
| Attention resolutions | 32, 16, 8 | 32, 16, 8 |
| Head Channels | 8 | 8 |
| Batch Size | 200 | 100 |
| Iterations | 6500 | 6500 |
| Embedding Dimension | - | 128 |

These configurations were manually tuned, and we did not observe significant differences when increasing their values. However, conducting a hyperparameter robustness study could be valuable for future work, as it may uncover optimal configurations that enhance model performance and stability.

## B.4 COMPUTE

The large models were trained on a machine equipped with an Nvidia L40 GPU (48 GB memory and 12 CPUs). Multi-session models required 30 hours of training, while multi-task models trained for 3 days, both completing a total of 6500 epochs. Single-session models were trained on a single Nvidia A40 GPU, with training times ranging from 4 to 8 hours depending on the session size. Session identification tasks required less than 30 minutes of training.

During inference, we observed high-quality sampling with 50 steps, without significant improvement when increasing the number of steps. The inference process takes less than a minute on a single GPU, or a few minutes on a CPU.

## B.5 MULTI-SESSION INPUT

Each neural population has a different number of neurons, which presents a challenge when working with multiple sessions. To address this, we standardize the input size by aligning all populations to the size of the largest population, padding the smaller populations. It is important to note that, similar to transformers, the padded portions of the input are ignored in the loss function during training, ensuring that the padding does not affect the model's learning process.

## B.6 DATA AUGMENTATION

To ensure our model remains invariant to the order of neurons, we implement a shuffling technique that randomizes the arrangement of neurons for each trial. This shuffling helps prevent the model from learning any spurious dependencies based on neuron positioning, thereby enhancing its generalization capabilities. While we did not investigate other augmentation techniques, we believe exploring them could be a promising avenue for enhancing the model's capabilities.

## B.7 OCCLUSION SENSITIVITY MEASUREMENTS

Partial occlusion studies are commonly used as a straightforward sanity check for models, allowing verification of the learning strategy in the input/output space. These studies help evaluate how sensitive the trained model is to occlusion. Traditionally, partial occlusion is applied to classification tasks, but we have extended this approach to multivariate forecasting.

In a given trial of neural activity, one neuron is left out while the rest are masked, and the predicted behavioral variables are recorded for each trial. For selective neurons—those showing significant predictability of behavior, such as for velocity along the x or y axis, or both—the analysis was refined with a 50 ms sliding window to determine the specific time window during which the neuron's activity encodes movement-related information.

# C   BENCHMARK DATASET

## C.1 DATASET COMPOSITION

In Table 3, the composition of our dataset suites is presented.

## C.2 DATA COLLECTION AND ORGANIZATION

The datasets integrated into our benchmark encompass a spectrum of tasks devised to evaluate various dimensions of motor control and primate behavior. All subjects featured in these datasets are *Rhesus macaques*.

Each task presents unique challenges and necessitates distinct forms of motor activity, thus furnishing a comprehensive framework for assessing the efficacy of neural decoding and forecasting models. The selection of these six datasets, based primarily on their higher quality compared to others, was motivated by factors such as data consistency, robustness of experimental design and wide-ranging behavioral and neural records, as well as comprehensive metadata. This ensures reliable and diverse data to effectively evaluate neural decoding and prediction models, based on data

Table 3: Composition of the Benchmark Dataset, detailing the tasks, number of subjects and sessions across various datasets.

| ID | Task | #Subj. | #Sess. | Brain Area | Ref. |
|---|---|---|---|---|---|
| 1 | RTT | 2 | 47 | M1, S1 | O'Doherty et al. (2017) |
| 2 | CO with Bump | 2 | 4 | Area 2 | Chowdhury et al. (2020b) |
| 2 | Two-Workspace RTT | 3 | 9 | | |
| 3 | Center-Out | 4 | 30 | M1 (Subj. 1&4) Area 2 (Subj. 2&3) PMd (Subj. 4) | Gallego-Carracedo et al. (2022) |
| 4 | CO | 2 | 23 | | |
| 4 | Wrist Isometric CO | 1 | 13 | M1 | Ma et al. (2023) |
| 4 | Key Grasp | 1 | 9 | | |
| 5 | CO / RTT | 4 | 117 | M1,PMd | Perich et al. (2024) |
| 6 | Maze | 2 | 9 | M1, PMd | Churchland et al. (2024) |

already used in a variety of previous studies. In addition, some data originates from the same source as the NLB, but we have extracted and transformed all datasets in full.

Initially obtained in disparate formats, including NWB and MATLAB files from distinct laboratories, we systematically mapped all datasets onto a standardized schema. This schema includes identifiers for the dataset, subject, task, session, and trial, thereby facilitating **efficient data filtering**. For instance, it is possible to efficiently filter data based on specific subjects, tasks, sessions, or trials of interest, streamlining analyses and facilitating comparisons across experiments.

**Dataset Standardization** Each dataset contains binned spike data, with units classified predominantly as single-units, save for dataset 1, where the initial unit of each session corresponds to multi-units (unsorted). The units of measurement within our dataset are standardized, with angles denoted in radians, hand position expressed in centimeters, hand velocity in centimeters per second, cursor position in millimeters, and cursor velocity in millimeters per second. Some datasets necessitated additional transformations; for instance, velocity information was computed offline based on position values where unavailable.

**Data Processing** Following processing, the datasets were converted into `Numpy` arrays and subsequently stored in `Parquet` format or NWB format. Notably, the decision not to standardize all datasets to a common bin size was deliberate, as it reflects another pertinent challenge encountered by foundation models. By retaining the original bin sizes across datasets, we provide a more authentic testing milieu for decoding and forecasting models, which often grapple with data granularity variability in real-world applications.

## C.3 TASKS DESCRIPTION

In this section, we describe each task. Figure 2 presents single trial plots for each task, including raster plots, behavior, and event indications.

**Center Out Target – CO** This task involves the animal repetitively reaching towards one of eight predefined targets upon receiving a cue, and subsequently returning to the center.

Monkeys performed the center-out (CO) reaching task using an upright planar manipulandum. Each trial began with the subject moving its hand to the center of the workspace. Following a random delay, one of eight peripheral targets was presented Gallego-Carracedo et al. (2022); Ma et al. (2023); Perich et al. (2024).

- Dataset 3: This dataset includes four subjects with recordings from different cortical areas, as detailed in Table 2.

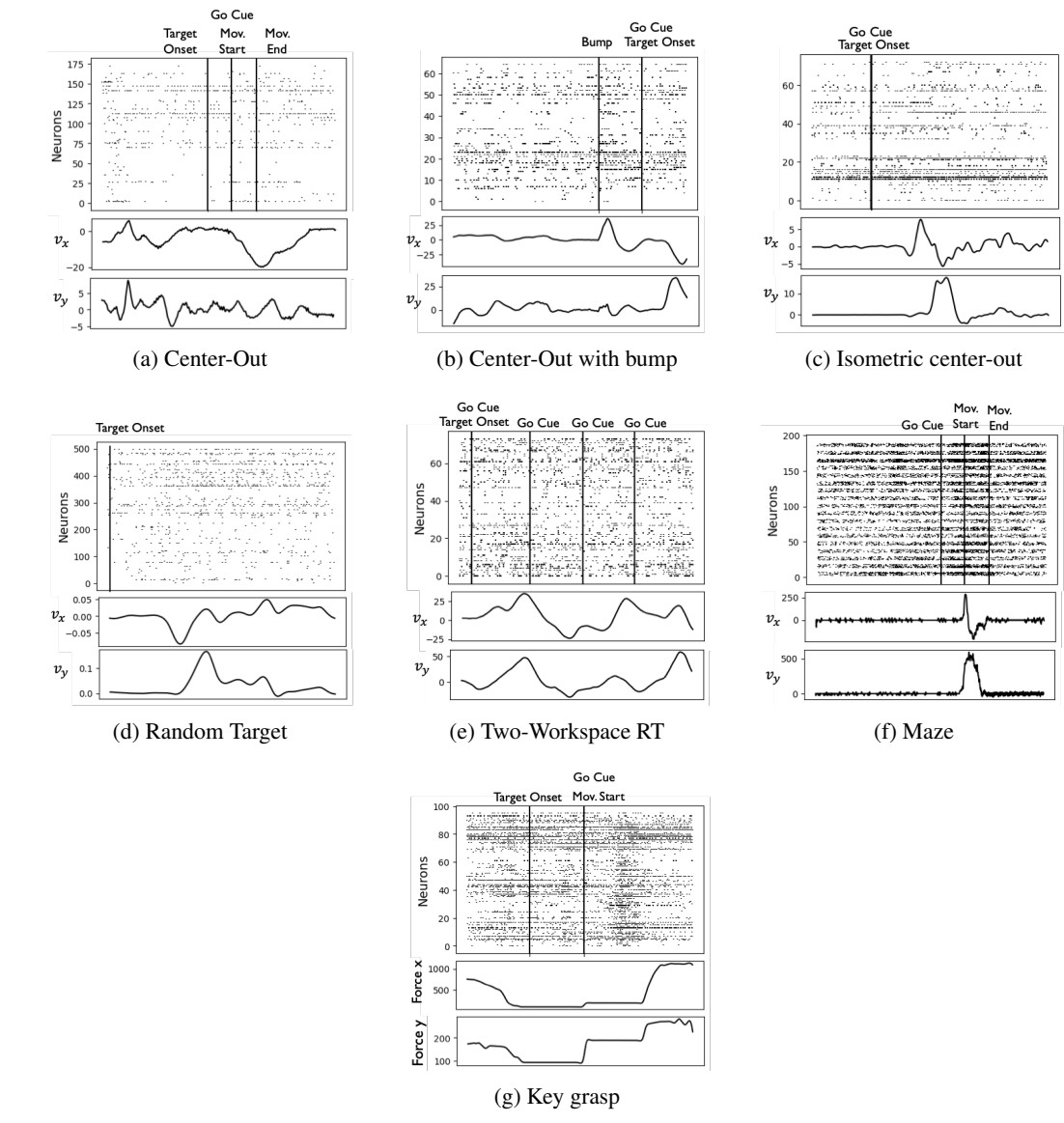

Figure 2: Benchmark dataset span seven diverse tasks. Sample spike rasters are aligned to task events.

(a) Center-Out    (b) Center-Out with bump    (c) Isometric center-out

(d) Random Target    (e) Two-Workspace RT    (f) Maze

(g) Key grasp

- Dataset 4: This dataset includes two subjects. The first subject performed the task with the right hand, while the second subject performed the task with the left hand Ma et al. (2023).
- Dataset 5: This dataset includes two subjects. For the first subject, the 'Go' cue was given immediately. For the second subject, there was a variable delay period of 500 – 1500 ms before an auditory 'Go' cue Dyer et al. (2017); Perich et al. (2024).

**Center Out Target with Bump – CO bump**    In this variation of the center-out target task, a perturbation or "bump" is introduced to the limb during the reach, requiring the animal to compensate for the disturbance while still reaching the target and returning to the center Chowdhury et al. (2020a).

**Wrist Isometric Center-Out Target – ISO**    The isometric wrist task involves monkeys controlling a cursor on a screen by exerting forces on a padded box around one hand. Different force directions move the cursor, requiring the monkeys to reach and hold targets for a liquid reward. Flexion and extension forces move the cursor right and left, while radial and ulnar deviation forces move it up

and down. Each trial starts with a center target hold, followed by one of eight outer targets and an auditory 'Go' cue. Monkeys must move the cursor to the target and hold it to receive a reward Ma et al. (2023).

**Maze** The Maze datasets consist of recordings from the primary motor and dorsal premotor cortices while a monkey performed reaches with an instructed delay to visually presented targets, navigating a virtual maze Churchland et al. (2010). The monkey completed various task configurations with differing target positions, numbers of virtual barriers, and barrier placements, leading to diverse straight and curved reach trajectories. Each configuration was attempted multiple times in random order, resulting in numerous trials per session.

The Maze datasets offers rich behavioral diversity, consistent performance across trials, and a large number of trials. This setup allows for averaging neuronal activity across trials, maintaining task variety to explore population activity Churchland et al. (2010); Pei et al. (2021); Gao et al. (2017). With an instructed delay paradigm, preparatory movements occur before the 'Go' cue, separating neural processes related to preparation and execution. This, along with the lack of unpredictable events, leads to predictable population activity during execution, resembling an autonomous dynamical system Churchland et al. (2012b); Shenoy et al. (2013); Churchland et al. (2012a); Pandarinath et al. (2018). These characteristics make the Maze datasets crucial in understanding neural population activity during movement preparation and execution Pei et al. (2021).

**Random Target – RTT** The random target task dataset contains motor cortical data from continuous, point-to-point reaches that start and end in various locations without delay periods and with highly variable lengths O'Doherty et al. (2017). This setup presents unique modeling challenges distinct from those posed by the Maze and Center Out datasets, where their stereotypy might constrain the complexity of observed neural signals Gao et al. (2017); Pei et al. (2021).

Given the absence of trial definitions in this task, we annotated the trial ID column, where each trial represents a different target, facilitating data segmentation for modeling. Consequently, the benchmark models fitted for this dataset were derived from random snippets of the continuous data stream. The unpredictability of these snippets, with new targets potentially appearing at any point within a data window, renders the simplification of autonomous dynamics a poor approximation Pandarinath et al. (2018); Keshtkaran et al. (2021).

**Two-Workspace Random Target – TRT** In this task, monkeys controlled a cursor using a two-link, planar manipulandum. The experiment involved reaching sequentially to visually presented targets in two distinct workspaces: one near the body on the contralateral side of the reaching arm, and one far from the body on the ipsilateral side. Before each trial, one of the two workspaces was randomly selected, and the monkey reached to a short sequence of randomly positioned targets within the chosen workspace Chowdhury et al. (2020a).

**Key Grasp** In the key grasp task, monkeys were trained to execute reaching and grasping movements towards a small rectangular cuboid gadget positioned beneath a screen using one hand. Cursor movements were controlled through force-sensitive resistors (FSRs) while aiming for targets displayed on the screen Ma et al. (2023).

The task required the monkey to perform a precise grasping action, involving the use of the thumb and index finger to grasp the cuboid gadget. The FSRs, located on the sides of the gadget, measured the applied grasping forces, with the sum and difference of their outputs determining the cursor's position on the vertical and horizontal axes, respectively Ma et al. (2023).

## D API FUNCTIONS

The benhcmark dataset has a minimalistic yet powerful API. To load data in our schema format, a single line of code suffices.

### D.1 NWB VERSION

```
from dataset_api import *
```

```
data = nap.load_file("sub-Animal-1-&-2.nwb")

# Retrieve data filtered to include only rewarded trials.
df, bin = get_dataframe(data, filter_result=[b'R'])
```

### D.2 PARQUET VERSION

```
from dataset_api import *

parquet_file_path = '2_10_Chowdhury_CObump.parquet'

# Retrieve data filtered to include only rewarded trials.
df, bin = load_and_filter_parquet(parquet_file_path, ['A', 'I', 'F'])
```

The following functions work the same for both versions. The `rebin` and/or `align_event` functions may be employed as required.

```
# Rebin the dataset with a bin size of 20 ms
df = rebin(df, prev_bin_size=bin, new_bin_size=20)

# Align each trial of the data (df) to a specific event ('
    EventTarget_Onset')
# The dataset has a bin size of 20 ms
# We want an offset of -20 ms before the event and 400 ms after the event
df = align_event(df, start_event='EventTarget_Onset', bin_size=20,
    offset_min=-20, offset_max=400)
```

## E DATASET DOCUMENTATION

### E.1 MOTIVATION

Q1. **For what purpose was the dataset created?**

A1. The purpose of this dataset is to provide a comprehensive benchmark for evaluating the accuracy, efficiency, and scalability of current and future multi-task, multi-session, and multi-subject models in large-scale scenarios. Currently, there is no benchmark dataset available for comparing these models. Additionally, this dataset aims to serve as an intermediate representation, bridging the gap between the metadata-rich and heterogeneous NWB/MATLAB files and machine learning algorithms. By unifying data from these diverse sources, this dataset is prepared and formatted for direct use in machine learning models.

### E.2 COMPOSITION

Q2. **What do the instances that comprise the dataset represent (e.g., documents, photos, people, countries)?**

A2. The dataset consists of three types of data:

– Neurophysiological data
– Behavior covariates
– Event indications

Q3. **How many instances are there in total (of each type, if appropriate)?**

A3. Our dataset includes a comprehensive collection of instances across multiple categories:

– 19 subjects
– 261 sessions

These instances are aggregated from six public datasets.

| ID | Task | #Subj. | #Sess. | #Neurons | #Trials | Brain Area |
|---|---|---|---|---|---|---|
| 1 | Random Target | 2 | 47 | 18406 | 25483 | M1, S1 |
| 2 | CO with Bump | 2 | 4 | 461 | 2766 | Area 2 |
| 2 | Two-Workspace | 3 | 9 | 629 | 4515 | |
| 3 | Center-Out | 4 | 30 | 1827 | 9226 | M1 (Subj. 1&4) Area 2 (Subj. 2&3) PMd (Subj. 4) |
| 4 | Center-Out | 2 | 23 | 2194 | 4712 | |
| 4 | Wrist Isometric CO | 1 | 13 | 899 | 2766 | M1 |
| 4 | Key Grasp | 1 | 9 | 864 | 903 | |
| 5 | Center-Out/Random Target | 4 | 117 | 11557 | 22317 | M1, PMd |
| 6 | Maze | 2 | 9 | 1728 | 23117 | M1, PMd |

Q4. **Does the dataset contain all possible instances or is it a sample (not necessarily random) of instances from a larger set?**

A4. The dataset is not a sample from larger sets; it is a curated collection of six entire datasets. These datasets were chosen for their high quality, data consistency, robust experimental design, and extensive behavioral and neural records, as well as comprehensive metadata. While we have preserved all essential data necessary for machine learning pipelines, we limited the metadata to streamline the datasets. The original datasets were rich in metadata, but we retained only the essential elements. Additional metadata and detailed descriptions will be made available on Kaggle (file descriptions).

Q5. **What data does each instance consist of?**

A5. Each instance in the dataset includes neurophysiological data, behavioral information, and event timings, detailed as follows:

**Neurophysiological Data:**

The columns for neurophysiological data are:

- **NeuronXX** (numeric): Represents single units for all datasets, except for dataset 1, where the first column per session corresponds to multi-units. Although we concatenated all the sessions, Neuron 1 in session 1 does not correspond to Neuron 1 in session 2. Each recorded population per session can be identified by dataset ID, animal, and session.

**Behavioral Data:**

The columns for behavioral data include:

- **target_dir** (numerical): Direction of the target in radians.
- **target_ID** (numerical): Identification of the target location. For example, in Center-Out tasks, there are 8 possible targets, each represented by an ID. **target_pos_x** and **target_pos_y** (numerical): Cartesian coordinates of the target position.
- **bump_dir** (numerical): Angle (in radians) of bump direction, if there was one. 0 radians is directly to the right, and $\pi/2$ radians is directly upward.
- **maze_num_target** (numerical): Number of targets (for the maze dataset).
- **maze_num_barriers** (numerical): Number of barriers in the maze.
- **force_x** and **force_y** (numerical): Interface forces between the hand and the manipulandum handle, in Newtons.
- **hand_pos_x** and **hand_pos_y** or **cursor_pos_x** and **cursor_pos_y** or **finger_pos_x** and **finger_pos_y**(numerical): Velocity of hand, cursor, or finger.
  **hand_vel_x** and **hand_vel_y** or **cursor_vel_x** and **cursor_vel_y** or **finger_vel_x** and **finger_vel_y** (numerical): hand, cursor or finger velocity.

**Events Data:**

The columns for events data are:

- **EventTarget_Onset** (boolean): Indicates when the target is presented.
- **EventGo_cue** (boolean): Indicates when the go cue is presented.
- **EventBump** (boolean): Indicates when there is a bump (only for Center-Out with bump task).

- **EventMovement_start** (boolean): Indicates when the subject starts moving.
- **EventMovement_end** (boolean): Indicates when the subject stops moving.

**Additional Information:**

We provide comprehensive indexes to efficiently filter the data by:

- **datasetID**: Identifier for each dataset (1 to 6)
- **animal**: Identifier for each animal in the dataset
- **session**: Identifier for each session of a particular animal
- **trial_id**: Identifier for each trial within a session performed by an animal from a specific dataset

We also provide indexes to filter data for rewarded trials and task information:

- **result** (categorical): Indicates the trial outcome: Aborted (A), Incomplete (I), Failed (F), Rewarded (R)
- **task** (categorical): Specifies the task name.

Q6. **Is there a label or target associated with each instance?**

A6. Yes, each instance can have associated labels or targets depending on the purpose of the model. For decoding models, all behavioral data covariates can be used as targets. For forecasting models, the data can be treated as a self-supervised learning task, using only the neurophysiological data.

Q7. **Is any information missing from individual instances?**

A7. There is no missing information from individual instances.

Q8. **Are relationships between individual instances made explicit (e.g., users' movie ratings, social network links)?**

A8. Yes, relationships between individual instances are made explicit. Each row in the dataset corresponds to a specific time point, ensuring that data across different columns and types (neurophysiological, behavioral, and events) are synchronized temporally. This alignment allows for clear and precise analysis of how different data points relate to each other over time.

Q9. **Are there any errors, sources of noise, or redundancies in the dataset?**

A9. Yes, since the data originates from publicly available experiments with animals, there are likely to be sources of errors and noise in the dataset. Experimental variability, biological factors, and environmental influences can all contribute to these imperfections. Our role was to curate and preprocess this data, not to collect it, so these inherent issues may persist.

Q10. **Is the dataset self-contained, or does it link to or otherwise rely on external resources (e.g., websites, tweets, other datasets)?**

A10. The dataset is self-contained. No links to external resources.

Q11. **Does the dataset contain data that might be considered confidential (e.g., data that is protected by legal privilege or by doctor-patient confidentiality, data that includes the content of individuals' non-public communications)?**

A11. There is no confidential data in this dataset.

Q12. **Does the dataset contain data that, if viewed directly, might be offensive, insulting, threatening, or might otherwise cause anxiety?**

A12. No.

Q13. **Does the dataset relate to people?**

A13. No.

Q14. **Does the dataset identify any subpopulations (e.g., by age, gender)?**

A14. No.

Q15. **Is it possible to identify individuals (i.e., one or more natural persons), either directly or indirectly (i.e., in combination with other data) from the dataset?**

A15. No.

Q16. **Does the dataset contain data that might be considered sensitive in any way (e.g., data that reveals racial or ethnic origins, sexual orientations, religious beliefs, political opinions or union memberships, or locations; financial or health data; biometric or genetic data; forms of government identification, such as social security numbers; criminal history)?**

A16. No.

### E.3 COLLECTION PROCESS

Q17. **What mechanisms or procedures were used to collect the data (e.g., hardware apparatus or sensor, manual human curation, software program, software API)?**

A17. Details about the data collection mechanisms and procedures can be found in the original papers cited earlier. These papers provide comprehensive descriptions of the hardware apparatus and other procedures used to collect the data.

Q18. **If the dataset is a sample from a larger set, what was the sampling strategy (e.g., deterministic, probabilistic with specific sampling probabilities)?**

A18. The dataset is not sampled; it comprises the entirety of the available data. Therefore, there is no specific sampling strategy involved.

Q19. **Who was involved in the data collection process (e.g., students, crowdworkers, contractors) and how were they compensated (e.g., how much were crowdworkers paid)?**

A19. The data collection process involved no direct participation or compensation, as the dataset consists of publicly available data.

Q20. **Over what timeframe was the data collected?**

A20. The data were collected in 2024 over a time period spanning six months.

Q21. **Were any ethical review processes conducted (e.g., by an institutional review board)?**

A21. No, such processes are unnecessary in our case.

### E.4 PREPROCESSING/CLEANING/LABELING

Q22. **Was any preprocessing/cleaning/labeling of the data done (e.g., discretization or bucketing, tokenization, part-of-speech tagging, SIFT feature extraction, removal of instances, processing of missing values)?**

A22. Yes, several preprocessing steps were performed on the data:

- Velocity Computation: In datasets lacking velocity information, velocity was computed based on position data. This step ensured consistency and completeness of the behavioral data.
- Neural Spiking Data Transformation: The neural spiking data, often provided in spike times, was transformed into spike counts with a bin size matching the behavioral data's time resolution. This transformation facilitated analysis and modeling by aligning neural activity with behavioral events.
- Event Alignment: All events assigned by experimentalists were matched to the final dataset, ensuring temporal alignment with other data. This step helped consolidate all relevant information into a single coherent dataset, facilitating subsequent analysis.

These preprocessing steps helped ensure data consistency, completeness, and alignment, making the dataset ready for analysis and modeling.

Additionally, we retained only the attributes essential for machine learning algorithms. Information related to electrode names, waveforms, and other such details available in the original datasets is not included in the current dataset.

Q23. **Was the "raw" data saved in addition to the preprocessed/cleaned/labeled data (e.g., to support unanticipated future uses)?**

A23. Yes, the raw data is available in the original repositories where the data was sourced from. This ensures that the original, unprocessed data is preserved and accessible for any future analyses or unanticipated uses.

Q24. **Is the software used to preprocess/clean/label the instances available?**

A24. No.

## E.5 USES

Q25. **Has the dataset been used for any tasks already?**

A25. The dataset was used only to generate the results available in the paper.

Q26. **Is there a repository that links to any or all papers or systems that use the dataset?**

A26. There are still no applications of the presented datasets. We intend to keep track of its uses in the project GitHub repo.

Q27. **What (other) tasks could the dataset be used for?**

A27. The dataset can be used to train encoding and decoding models.

Q28. **Is there anything about the composition of the dataset or the way it was collected and preprocessed/cleaned/labeled that might impact future uses?**

A28. We believe that our dataset will not encounter usage limit.

Q29. **Are there tasks for which the dataset should not be used?**

A29. No, users could use our dataset in any task as long as it does not violate laws.

## E.6 DISTRIBUTION

Q30. **Will the dataset be distributed to third parties outside of the entity (e.g., company, institution, organization) on behalf of which the dataset was created?**

A30. Yes, the dataset will be made publicly accessible.

Q31. **How will the dataset will be distributed (e.g., tarball on website, API, GitHub)?**

A31. It will be distributed on Kaggle and Dandi Archive.

Q32. **When will the dataset be distributed?**

A32. After the review.

Q33. **Will the dataset be distributed under a copyright or other intellectual property (IP) license, and/or under applicable terms of use (ToU)?**

A33. The dataset is licensed under the Creative Commons CC BY-NC-ND 4.0 license.

Q34. **Have any third parties imposed IP-based or other restrictions on the data associated with the instances?**

A34. No.

Q35. **Do any export controls or other regulatory restrictions apply to the dataset or to individual instances?**

A35. No.

## E.7 MAINTENANCE

Q36. **Who is supporting/hosting/maintaining the dataset?**

A36. The authors of the paper.

Q37. **Is there an erratum?**

A37. No, there is no erratum as of yet. If necessary in the future, an erratum will be developed for the dataset, as well as for this document.

Q38. **Will the dataset be updated (e.g., to correct labeling errors, add new instances, delete instances)?**

A38. There are no current plans on updating the current datasets. This can change in the future, either to introduce new variants to the dataset, or to correct any undetected bug.

Q39. **If the dataset relates to people, are there applicable limits on the retention of the data associated with the instances (e.g., were individuals in question told that their data would be retained for a fixed period of time and then deleted)?**

A39. There are no applicable retention limits of the data.

Q40. **Will older versions of the dataset continue to be supported/hosted/maintained?**

A40. If any updates are published, previous versions will be available.