# OpenReview forum: "One Model to Train Them All: A Unified Diffusion Framework for Multi-Context Neural Population Forecasting"
_ICLR.cc/2025/Conference — Submitted to ICLR 2025_

### Official Review · Reviewer_hXMN · 2024-10-28

**Soundness:** 3
**Presentation:** 3
**Contribution:** 3
**Rating:** 6
**Confidence:** 4

**Summary:**

In this paper, the authors present a joint neural spiking activity and behavior forecasting model based on diffusion mechanisms. They demonstrate that the novel architecture can jointly encode both neural activity and behavior, and that trainign can be done on multiple animals, tasks, and sessions. The authors also introduce a novel dataset which incorporates data from non-human primates from several labs, which is curated for easy model development. Finally, the authors show that their new model can easily forecast behavior, and neural activity, from new sessions with no fine tuning.

**Strengths:**

This paper is one of (if not the first) account of diffusion modelling for neural data and bahvioral forecasting. The development of this application for diffusion modelling is an important strenght, and a good contribution to the existing litterature. Furthernmore, the introduction of a novel format for datasets suited for general multi-session, multi-lab modelling is a great contribution. Finally, the results presented are encouraging.

**Weaknesses:**

As indicated above, I do believe the model and approach presented in this paper is an important contribution. However, I do think there are weaknesses with respect to the results presented, as well as clarity issues in the presentation along with some mismatch with the claims put forth. See more details in the specific questions below.

**Questions:**

- No comparison of neural activity forecasting with other models:
While the authors clima throughout this paper the their model is a significant advance in joint forecasting of behavior and neural activity, as far as I can tell, only Fig.3C shows forecasting of neural activity, and it only compares prediction across two variants of the proposed model. While there is significant benchmarking of behavioral forecasting, can the authors explain why they report such minimal experiments on neural activity forecasting while putting the two forecasting modalities on equal footing in their claims?

- No comparison to behavior decoding with other foundation models:
For example, the POYO model from Azabou et al., often cited in this manuscript, does behavior forecasting on 500ms windows (as in this paper) on similar cursor control tasks but achieves R^2 of around 0.95+. This is in contrast with the presented results which seem in the 0.6 range. I do not think the present model should be SOTA on forecasting to warrant publication, but it should acknowledge comparative performance of other models, and explain clearly the differences in the approaches that explains this difference. Can the authors explain these points?

- Confounding source of forecastinginformation:
As noted above, most of the forecasting results focus on behavior (i.e. cursor velocity). However, cursor velocity is also present in the input data, and forecasting happens on joint cursor veloity and spiking activity representations. Have the authors tried forecasting behavior only from behavior, only from spiking activity? What is the effect of the regularization terms in the loss that weight both of these modalities? Often times, models present with multiple sources of data and that aim to forecast one of them can rely heavily on one source rather than the combined effect of two. It is important to understand the behavior of the model under multiple sources.

- Potential ablations:
I realize that getting ablation requests from reviewers is frustrating. However, this model relies on important prprocessing that includes spike train binning followed by Gaussian smoothing. Have the authors tries different binning and smoothing resolutions? Ho robust is the model for variation on these steps?

Minor:

- Captions often indicate R^2 forecasting error but it is not indicated what quantity is being forecasted. Is it cursor velocity (I believe this is it in most cases) or spiking activity? Captions should be precise on this front.

- Line 220: authors indicate "$\mathcal{D}$-dimensional [...]" but here D is a dataset. Not sure what is meant there or if this is a typo.

- Line 235: "[...} have been shown to be better for [...]" citation needed.

- In C4, the authors mention the presence of unconditional trials in training. It is not clear what this means in this context. More details about what trial information is removed, or uncondtiioned on, is needed.

- Related work, "Multi-session Training and Alignment". The authors review past work that aim to do transfer learning for multi-session inference. However they do not acknowledge that the other efforts they cite under the "Foundation models for neuroscience" section also achieve this. It is important to acknowledge that recent work does achieve borad generalization across sessions, animals, labs, etc. Even is the authors do acnkowldge these past results later on, they cast them as "specifically designed for decoding brain-machine interfaces" which is not accurate. If anything, these past works work on very simlar objectives and datasets as the current work.

I would be happy to revise my score if the authors provide adequate answers to these questions.

---

> ### Author Response · Authors · 2024-11-18
>
> - Questions:
>
> 1. Thank you for the great question. Typically, models focused on forecasting primarily report behavior forecasting results, as it serves as an easier and more direct metric. In these cases, behavior forecasting often reflects how well the model forecasts neural activity, especially since those models are not jointly trained. Therefore, behavior forecasting becomes a proxy for neural activity forecasting performance. This makes it challenging to find direct comparisons, particularly because our model uses smoothed spikes, whereas models like SeqVAE, which also use smoothed spikes, do not report neural activity values.
> However, we acknowledge the lack of results regarding neural activity and have added relevant plots in Fig. 3.E to address this gap.
>
> 2. We would like to clarify that POYO is not capable of forecasting; instead, it performs filtering, reporting results in 500 ms windows. This means it predicts the next time step of behavior variables based on a history window of neural activity and slides this history window until the desired length of predicted behavior is achieved. To the best of our knowledge, ours is the only unified model designed for forecasting.
> We compared the performance of our model with the most recent state-of-the-art models: XFADS (for single-session) and SeqVae (for multi-session). However, based on your feedback, we also adapted our model to perform filtering to evaluate its performance in this context. The results show that our model is as competitive as NDT2 and POYO, despite filtering not being its primary focus. The design choices made for our model were not optimized for filtering, as diffusion models are traditionally better suited for full-sequence generation rather than next-token prediction, as is the case with transformers.
> The new experiments have been added to the manuscript in Fig. 3.C and Fig. 5.D. A more detailed comparison between our approach and other models can be found in Appendix A.
>
> 3. These are very relevant questions: (1) Yes, we tested this and have added the results to Fig. 3.B. Additionally, we conducted an ablation study comparing the model's performance with and without behavioral data (Fig. 6, first plot). We found that including or excluding behavioral data did not impact the model’s performance in terms of neural activity forecasting. However, we would like to clarify that the difference observed in behavioral performance stems from the fact that, when trained solely on spiking data, we need to use a linear decoder (ridge regression in this case) to map the predicted neural activity to behavior values. As a result, this performance metric reflects the quality of neural activity prediction, not the model’s ability to forecast behavior directly.
> (2) We thank the reviewer for the suggestion to conduct a lambda ablation study. We trained the model with different lambda values and reported the performance in terms of both neural spiking activity and behavior. The lambda term in the loss function significantly influences model performance. Our benchmarks indicated that a value of 0.5 worked well for both behavior and spiking activity. However, this value may vary depending on the specific brain area or task being studied (Fig. 6, last plot). This dependency is expected, as increasing the attention the model gives to behavior can reduce its accuracy in predicting neural activity, since not all neurons are behavior-relevant. Conversely, focusing more on neural activity can decrease behavior forecasting accuracy.
> (3) We do not believe that our model excessively focuses on one modality, especially in this case. For the forecasting results, we followed the preprocessing strategy used by the models we compared against, which involved aligning the data to movement onset. This means that, in the history provided to the model for forecasting what happens after movement onset, the behavior variables are simply set to 0’s. As a result, the model is not biased toward behavior but instead relies on the neural activity to make predictions after the onset.
>
> 4. Once again, we thank the reviewer for the suggestion to run ablation studies. We conducted several experiments with different time bins, fixing the smoothing filter, as well as with fixed time bins and varying smoothing filters. Our results show that the model is robust to time bins larger than 5 and across different smoothing filters. The choices of both the time bin and smoothing filter used in the paper were based on those applied by our baseline models.

---

> > ### Author Response · Authors · 2024-11-18
> >
> > - Minors:
> >
> > We thank the reviewer for pointing out the typos and the lack of detail in certain sections. We believe we have addressed all of these issues in the updated version of the manuscript. Specifically, we clarified the quantity being forecasted, corrected the typos, added details on the unconditional/shared session aspect, and changed the title to "Multi-session Forecasting". This new title reflects that all of the methods presented forecast using multiple sessions, whereas existing "Foundation models" were designed primarily for filtering, which is more relevant for brain-machine interfaces.
> > We also agree that both groups of approaches aim to train with data from multiple sessions. We hope that the distinction we intended to make is now clearer.
> >
> > We hope the reviewer found our rebuttal helpful. If there is anything we can further clarify or do to improve the reviewer's overall assessment of our work, please let us know.

---

> > ### Comment · Reviewer_hXMN · 2024-11-21
> >
> > I thank the authors for the detailed reply and for the notable extra work to address my concerns. Steps taken address my concerns for questions 2 and 4. I have remaining questions for 1 and 3.
> >
> > 1. Claims of neural forecasting. I am not certain i agree with the authors when they state that "behavior forecasting becomes a proxy for neural activity forecasting performance". In the paper, lines 73 to 95, the authors state that "we enable zero-shot learning for forecasting neural data, thereby effectively addressing the limitations associated with traditional few-shot learning approaches". However, in the rebuttal answer, the authors argue that typically models just forecast behavior. A number of models can forecast neural activity, either in the form of (smoothed) spikes, firing rates, or other. The claim made in this paper is that the proposed model can jointly forecast neural activity and bahavior, but the authors seem to want to avoid comparison of direct neural activity forecasting by saying behavior is a good proxy for neural activity. This argument seems a bit disengenuous. The addition of panel E to figure 3 illustrates what forecasting is, but there is still no reported performance quanity for neural forecasting, and no comparison to other methods. I know that such quantities are now reported in later figures, but only in ablation contexts. While I understand that running several benchmarking experiments is not feasible, I think that either the claims of neural forecasting should be revised/qualified (as there are no comparisons) or comparisons should be presented.
> >
> > 3. I thank the authors for producing a new plot in figure 3, this is indeed revelaing. However I find newly added Figure 6 intriguing but difficult to interpret as the caption does not explain what is presented. If I understand the first panel of Figure 6, it seems like bahavior prediction for the model trained on both spikes+bahvior performs worst than the model trained on spikes alone. This is very confusing, unless I am not reading this right. Can the authors explain why this is the case? It seems to contradict the claim tha joint behavior and neural activity forecasting is beneficial. Also, could the authors provide information about what is presented in the other panels of fig 6?
> >
> > I wish to thank the authors for the excellent work and a very nice model architecture. I recognize the efforts put into improving the paper. I am now adjusting my score to a 6 and I remain attentive for replies to my points above.

---

> ### Author Response · Authors · 2024-11-22
>
> We thank the reviewer for acknowledging our efforts to improve the manuscript and for the updated score. Below, we address the remaining questions:
>
> 1. We thank the reviewer for highlighting the importance of comparing both behavior and neural forecasting, a point we initially thought was less critical. We also appreciate the reminder regarding the existence of several studies that report these values, which we had overlooked. In response, we have now included the reporting values for single-session baselines for neural activity forecasting and next-step prediction (Fig. 3B and C in the new version), alongside the previously reported behavior forecasting results.
>
>
>
> 2. We apologize for the unclear reporting of the ablation study results and thank the reviewer for pointing this out. We have now added a description for each panel in Fig. 6. Regarding the spikes vs. spikes+behavior results, we appreciate the reviewer’s feedback on the confusion. The discrepancy arose because we reported results for forecasting over a 500ms window, where the better performance of the spikes-only model was due to ridge regression compensating for forecasting errors in spike predictions, leading to improved behavior accuracy. To address this, we have updated the results in the new version to reflect the filtering mode of the model. Panel A of Fig. 6 (in the updated version) compares our joint model with the two-step model, where behavior is predicted using ridge regression based on next-step spiking activity. The results show that the joint model outperforms the two-step model in behavior prediction while preserving spiking activity accuracy. This demonstrates that the joint model effectively integrates spiking and behavioral variables, optimizing both simultaneously. Unlike the two-step model, which treats these tasks separately, the joint model better captures their interactions, resulting in improved performance without sacrificing spike prediction accuracy.
>
> Panels B-E show the effects of time bin size, smoothing filter, history window, and forecasting window on $R^2$  for behavior forecasting. Specifically, for each variable (e.g., time bin size), we evaluated the model's behavior performance while keeping all other variables constant.
>
> - **Panel D**: We examined the behavior accuracy using different history forecasting windows: 700 ms, 500 ms, 200 ms, and 100 ms. As expected, we observed a significant performance drop at 100 ms, which aligns with our finding that the most relevant neural activity for behavior occurs 200-250 ms before movement onset.
>
> - **Panel E**: We explored the impact of different forecasting windows on model performance, assessing how the forecast horizon influences behavior prediction.
>
> - **Panel F**: We trained the model with three different lambda values and reported the resulting behavior and spiking activity forecasting performance.

---

> > ### Author Response · Authors · 2024-11-25
> > **New experiments and comparisons**
> >
> > We would like to highlight that we conducted additional experiments to emphasize the differences between our proposed approach and previous unified models.
> >
> > We ran new experiments to address the previous concerns:
> >
> >   **Diffusion Training:**  To test whether the improved performance is due to our architecture or the training strategy, we trained the model without it (details in Appendix B.2). Our results show that the architecture itself is inherently superior, and that training with diffusion further enhances its performance.
> >
> > | Model  | $R^2$ (hand velocity) 500 ms forecasting |
> > |------------------|-----------------|
> > | LFADS  |   0.34 ± 0.05 |
> > | DVBF   |  0.33 ± 0.04|
> > | XFADS   |  0.74 ± 0.04 |
> > | NDT2  |0.30 ± 0.08  |
> > | Ours - diffusion training |  0.78 ± 0.05|
> > | Ours - without diffusion training |  0.55 ± 0.04|
> >
> > The poor performance of NDT2 with a causal mask was expected, since recent work on transformer architectures has shown that learning global dependencies in temporal tokens leads to poor prediction results on multivariate time series datasets [1]. Thus, this model is not competitive in terms of forecasting and, in fact, that was not its original purpose, nor does POYO's architecture allow for forecasting, so our model is the only unified model proven to be able to so competitively with state-of-the-art neural data forecasting models.
> >
> > In Fig.1 of the Appendix B.6 we show that training without diffusion required substantially more training iterations and consistently yielded lower prediction accuracy. These findings align with previous studies emphasizing the importance of intermediate noise steps in diffusion training for enhancing model prediction quality [2].
> >
> >  **Contribution:** We would like to emphasize that our primary goal is forecasting, and previous unified models either cannot perform forecasting or their performance is not competitive with single-session neural data forecasting. To address this limitation, we have developed a distinct architecture and training strategy. Our model is specifically designed to handle both long-term and short-term dependencies for accurate forecasting. Additionally, we improved the performance and training efficiency through the use of diffusion.
> >
> > Given these updates, we hope the revisions address the concerns raised and provide a clearer demonstration of the contributions and significance of our work.
> >
> > [1] Liu, Y., Hu, T., Zhang, H., Wu, H., Wang, S., Ma, L., & Long, M. (2023). itransformer: Inverted transformers are effective for time series forecasting. arXiv preprint arXiv:2310.06625.
> >
> > [2] Nichol, A. Q., & Dhariwal, P. (2021, July). Improved denoising diffusion probabilistic models. In International conference on machine learning (pp. 8162-8171). PMLR.

---

> > > ### Comment · Reviewer_hXMN · 2024-11-28
> > >
> > > I thank the authors for the hard work and continued attention to review points. This review process has been demanding and the authors have risen to the challenge.
> > >
> > > I welcome the changes made in light of my latest replies. There is one thing that still bothers me, however. As the authors say themselves, their main focus is on forecasting, and they claim that jointly training on spiking and behavior enhances forecasting in both modalities. However, after I pointed out a strange result in figure 6 that showed that forecasting seemed to do worst in models that were jointly trained, the authors amended Figure 6 to show results on filtering instead of forecasting. In this new figure, the advantage of joint training is also minimal, but present.
> > >
> > > In the end, I do believe this paper is good enough for publication, but there are some inconsistencies in the message that should be adressed. I understand that the pdf can no longer be updated, and I am not requesting more experiments. However I would be curious to hear the authors on the point I am making above. I also intend to actively engage with the AC and the other reviewers should this paper be considred as a borderline case.

---

> > > > ### Author Response · Authors · 2024-11-29
> > > >
> > > > We thank the reviewer for the thoughtful and encouraging comments. We deeply appreciate the time and effort invested in this review process and are grateful for the valuable feedback, which has significantly contributed to improving our work.
> > > >
> > > > We would like to clarify that our main goal is not to claim superior performance with joint training compared to a two-stage approach. Instead, the primary motivation behind joint training is to address the assumption made by the two-stage approach and to learn low-dimensional latents that are highly predictive of behavior without sacrificing their fit to the neural data. This was the goal of our ablation study.
> > > >
> > > > In the dataset we tested, which consists of M1 recordings, neural activity is known to exhibit a near-linear relationship with motor outputs. Therefore, it is unsurprising that the performance improvement from joint training was minimal compared to the two-stage approach with ridge regression (aligned with [1]).
> > > >
> > > > That said, joint training offers additional benefits beyond performance, such as enabling the identification of neurons most important for behavior. We acknowledge that this could be made clearer in the manuscript and are happy to revise the text accordingly.
> > > >
> > > > Although the reviewer did not request additional experiments, we were curious to explore whether joint training provides performance gains in datasets where the relationship between neural activity and behavior might be more nonlinear. To investigate this, we tested it on Dataset 3 (Subject 4), which includes recordings from PMd. The results are shown below:
> > > >
> > > > | Model  | Prediction | 500 ms forecasting |
> > > > |------------------|-----------------|-----------------|
> > > > | Spikes + Linear | neural activity |   0.68 ± 0.1 |
> > > > |  Spikes + Linear | behavior   |  0.71 ± 0.04|
> > > > |  Spikes + behavior | neural activity   |  0.67 ± 0.08 |
> > > > | Spikes + behavior | behavior |0.82 ± 0.06  |
> > > >
> > > > In this case, joint training improved performance for behavior prediction while maintaining neural activity prediction accuracy. Unlike the M1 dataset, where the relationship between neural activity and behavior was near-linear, the PMd dataset appears to benefit from the ability of joint training to handle nonlinear dependencies.
> > > > This suggests that in the previous case, the near-linear relationship allowed the two-stage approach to perform well, and any potential gains from joint training were blurred by the error compensation inherent in the two-stage forecasting process.
> > > >
> > > > These results demonstrate that joint training can yield significant improvements in more complex datasets, where nonlinearities between neural activity and behavior are more pronounced, which aligns with [2].
> > > >
> > > > [1] Hurwitz, C., Srivastava, A., Xu, K., Jude, J., Perich, M., Miller, L., & Hennig, M. (2021). Targeted neural dynamical modeling. Advances in Neural Information Processing Systems, 34, 29379-29392.
> > > >
> > > > [2] Sani, Omid & Pesaran, Bijan & Shanechi, Maryam. (2024). Dissociative and prioritized modeling of behaviorally relevant neural dynamics using recurrent neural networks. Nature Neuroscience. 27. 2033-2045. 10.1038/s41593-024-01731-2.

---

> > > > > ### Comment · Reviewer_hXMN · 2024-12-02
> > > > >
> > > > > I thank the authors for their dedicated and continued work. I appreciate the additional experiments presented in the last comment. However, I am perlexed to see that joint spike+behavior training yields worst prediction than training on spikes alone. I understand that the notable result is that behavior forecasting behaves better in the joint training scheme, although this is not the case across all modalities.
> > > > >
> > > > > A frew items:
> > > > >
> > > > > 1-  The authors say :"We would like to clarify that our main goal is not to claim superior performance with joint training compared to a two-stage approach. Instead, the primary motivation behind joint training is to address the assumption made by the two-stage approach and to learn low-dimensional latents that are highly predictive of behavior without sacrificing their fit to the neural data."
> > > > >
> > > > > I am not certain how this statement differs from saying that jointly-trained models are better for neural data modelling, which is in essence what the paper proposes.
> > > > >
> > > > > 2- There is still some questions between filtering vs forecasting results. In the revised Fig 6., tha authors state
> > > > > "[...] we have updated the results in the new version [of Fig 6] to reflect the filtering mode of the model. Panel A of Fig. 6 (in the updated version) compares our joint model with the two-step model, where behavior is predicted using ridge regression based on next-step spiking activity."
> > > > >
> > > > > I am a bit bothere that the rest of the paper still focuses on forecasting but now only this figure reports filtering since the forecasting result is not aligned the message. The authors explain the reason why this might be the case in their comment above, but this explanation remains speculative and is not well established in the text. Indeed, the caption of Fig 6 states that the results shown are for forecasting while the authors say in this forum that the they are for filtering. Because of this discrepency and the need to better understand the effect of joint training, I will keep my score.
> > > > >
> > > > > I remain available to discuss with fellow reviewers and I still believe that this paper could be published given at the very least that the discrepencies I outline above are made transparent in the text.

---

> ### Author Response · Authors · 2024-12-02
>
> We appreciate the reviewer’s feedback and would like to address the concerns raised.
>
> 1. **Fig. 6 and Filtering vs. Forecasting:**
> It is not accurate to state that Fig. 6 shows filtering results in all panels. While panel A focuses on filtering, the remaining panels explicitly focus on forecasting, as noted in the manuscript. This approach aligns with the discussion in lines 512–520 of the manuscript. Filtering, equivalent to one-step-ahead forecasting, minimizes error propagation and isolates the model’s ability to predict neural activity accurately. We chose to highlight one-step forecasting results (20 ms) rather than longer windows (e.g., 500 ms) because, over extended horizons, the ridge regression in the two-stage approach compensates for neural forecasting errors. Ridge regression decodes behavior from already forecasted neural activity, rather than directly forecasting behavior from prior neural activity. This creates an unfair comparison, as ridge regression smooths over the limitations of the two-stage model. The degradation in forecasting accuracy with longer windows is evident in panel E and is expected. We could add a figure to further illustrate this phenomenon, showing how the two-stage method's performance remains relatively stable due to ridge regression, while the joint model performance decreases in line with neural forecasting degradation, with the increase of the forecasting window. However, we emphasize that Fig. 6 remains aligned with the paper's primary focus on forecasting and the evaluation of joint training’s benefits.
>
> We recognize that the M1 dataset used in Fig. 6 may not optimally highlight performance improvements with joint training. The goal of joint training is not solely to outperform the two-stage approach but to address its assumptions, learning low-dimensional latents that are predictive of behavior while preserving neural fit. While performance gains may appear minimal in datasets with near-linear dependencies (e.g., M1), joint training excels in interpretability and is particularly effective in modeling nonlinear dependencies, as demonstrated in the PMd experiments.
>
>
> We hope that our clarifications address the reviewer’s observation that "joint spike+behavior training yields worse prediction than training on spikes alone." Our results suggest joint training preserves neural fit and improves behavior forecasting in some scenarios, with weaker performance only observed in certain datasets (e.g., M1) when ridge regression compensates for long forecasting windows. However, we observed benefits in shorter windows.  We would greatly appreciate it if you could consider revisiting your evaluation in light of these clarifications.

---

> > ### Comment · Reviewer_hXMN · 2024-12-03
> >
> > Fig 6 Panel A shows results for filtering, not forecasting. The rest of the panels do indeed present results for forecasting. This distinction is not well indicated in the caption. Furthermore, text starting at line 512 states:
> >
> > "
> > Panel A of Fig. 6 provides a direct comparison between our joint model and the two-step model,
> > where behavior is predicted using ridge regression based on the next-step spiking activity prediction
> > (filtering). The results show that the joint model outperforms the two-step model in behavior predic-
> > tion, while maintaining the quality of spiking activity prediction. Thus the joint model effectively
> > learns the relationship between spiking and behavior in an integrated manner, optimizing for both
> > simultaneously. Unlike the two-step model, which separates the tasks of spikes and behavior predic-
> > tion, the joint model captures the interactions between neural activity and behavior more effectively,
> > leading to improved overall performance. Importantly, the joint model does not sacrifice the accu-
> > racy of spike predictions, suggesting that it can handle multiple objectives without degradation.
> > "
> >
> > Nowhere in this text is it indicated that this advantage of joint modelling is only valid for filtering, and actually is detrimental for forecasting, as originally shown in the first revised pdf. As the rest of the paper treats forecasting, this line of argument is sloppy at best, and misleading at worse. I understand that the authors have arguments related to the nature of the dataset which support the reason why filtering works best in non-joint training, and that this trend might reverse with other datasets. Nevertheless, the current text does not communicate this adequately, and one might argue that more experiments are needed on other datasets. I am willing to support the paper but clarity of results is paramount. I stand by my current score.

---

> > > ### Author Response · Authors · 2024-12-03
> > >
> > > In Fig. 6A, the label ‘Filtering’ is explicitly indicated in the figure itself, and the captions of the other panels clearly refer to ‘Forecasting’. We think this distinction is clear. But we can change it at the suggestion of the reviewer.
> > >
> > > We appreciate the reviewer’s feedback and agree this section could be clearer. The key advantage of joint training is its ability to learn neural activity and behavior dynamics simultaneously, as shown throughout the paper where the model performs well on both tasks in all scenarios (filtering and forecasting). Thus training the model with a 2-step approach does not allow us to forecast behavior (in the true meaning of it -  it is only smoothing the forecasted neural activity to behaviour, the model does not learn the behavior dynamics).
> > >
> > > However, we propose that the quoted text be clarified as follows:
> > >
> > > ‘Panel A of Fig. 6 shows a comparison between joint training of the model and two-stage training, where in the latter case the neural activity is smoothed to predict behaviour using ridge regression, which is not the same as actually forecasting behaviour. Thus, joint training is the only way to make the model learn the dynamics of both, which is the reason for choosing this type of training.
> > >
> > > Furthermore, the results show that the joint model does not sacrifice the accuracy of spike predictions, suggesting that it can handle both modalities without degradation.
> > >
> > > Unlike the two-stage model, the joint model captures the interactions between neural activity and behaviour more effectively, since it is not subject to the linear assumption. Furthermore, the joint model has the potential to improve overall performance in scenarios where the linear assumptions of the two-step approach deviate from reality, and do not in the case of a near-linear relationship. ’

---

### Official Review · Reviewer_SWNx · 2024-11-03

**Soundness:** 3
**Presentation:** 3
**Contribution:** 2
**Rating:** 6
**Confidence:** 4

**Summary:**

This work proposed a foundation model for solving adaptation in multi-animal, multi-session, multi-task for neural dynamics, including both dynamical forecasting and decoding tasks. It demonstrated its performance on six electrophysiology datasets from monkeys with recordings from different brain areas and different BCI tasks. It utilized a conditional diffusion models to perform unsupervised alignment to transfer unseen data aligned with original dataset, and also a session embedding to compensate for session variability. The model is trained with a causal forecasting tasks with self-supervised tasks. It learns a shared latent representation with both behavior and neural activity. It evaluated on both single-session and multi-sessions with a session identification approach or a zero-shot learning.

**Strengths:**

1. This paper is proposed to address a well motivated and important question in neural data analysis, and focus on achieving generalization across multi-animal, multi-session, multi-tasks for neural dynamics forecasting and behavior decoding.
2. This paper is evaluated on extensive experiments including 6 datasets, 19 monkeys with enhanced predictive performance compared with existing approaches.
3. Extensive techniques such as conditional diffusion model, meta-learning, session-identification are exploited to address the problems.
4. The work is developed with great accessibility including introducing a benchmark dataset and APIs to benefit the broader community.

**Weaknesses:**

1. Existing works like NDT2 and POYO also focused on developing a unified framework (foundation model) for neuroscience. Although this work claims to integrate with additional forecasting capabilities, the core of this approach does not deviate a lot from those existing frameworks, for example, this work also utilized the session identification techniques proposed in POYO and demonstrate the effectiveness of this approach.
2. More systematic investigation about the limitations about the forecasting capability is necessary, what is the maximal time length of effective forecasting, how much historical information is needed, how does the model perform to address the trial-to-trial variability?
3. This work integrate with both behavior data and neural data to learn a joint representation space, it would be helpful to perform an ablation study to explain how behavior information is useful or without behavior information, will this method still be effective for forecasting or behavior decoding?

**Questions:**

1. Any potential hypothesis or assumptions about why zero-shot learning outperforms session identification while session identification is better when more data is available?
2. Add more clarification for the notations in C.1 and C.2, are time steps $t$ in neural activities the same as the notation $t$ in diffusion processing?
3. Clarify the major contributions and novelty compared to existing NDT2 and POYO models.
4. Time complexity of the proposed approaches compared with existing baselines.

---

> ### Author Response · Authors · 2024-11-18
>
> - Weaknesses:
>
> 1. Yes, our goal is forecasting, not filtering, as is the case with NDT2 and POYO. We acknowledge that the original manuscript did not clearly distinguish our approach from these two models. To address this, we have added a detailed comparison in Appendix A to highlight the differences between our model and NDT2/POYO.
>
> 2.	We thank the reviewer for the suggestion to systematic explore the limitations of forecasting capabilities of the model. We ran several ablation studies exploring different history forecasting windows and forecasting windows, with the results presented in Figure 6 of the updated manuscript.
> For panel D, we saw what is the behavior accuracy using a history forecating window of 700 ms, 500 , 200 and 100 ms. The model showed a huge performance degradation for the 100 ms what was expected due to the fact that we found that the most revelant neural activity for  behavior was presnet in 200- 250 ms before the movement onset. For E, we explored how good is the forecating performance for different forecating windows.
>
> 3. We conducted the recommended ablation study to evaluate the model's performance with and without behavioral data (Fig. 6, first plot).
> The results show that the joint model outperforms the two-step model in behavior prediction while preserving spiking activity accuracy. This demonstrates that the joint model effectively integrates spiking and behavior variables, optimizing both simultaneously. Unlike the two-step model, which separates these tasks, the joint model better captures their interactions, leading to improved performance without sacrificing spike prediction accuracy.
>
> - Questions:
>
> 1. Thank you for the insightful question, which helps clarify our results. In the new version, we added our explanation/hypothesis: Our transfer strategies aim to learn shared features while capturing session-specific styles via embeddings. Training on one small session, however, is insufficient to build a noise-robust model, and few-shot examples from another animal fail to separate shared and session-specific features, pushing the model further from the data manifold and amplifying noise in the learned representation.  With sufficient data, the model learns robust shared features, and few-shot examples capture session-specific styles, improving performance. This emphasizes the importance of adequate data for robust representations. While Azabou et al. (2023) showed that large-scale pretraining enables session identification to adapt effectively, our results highlight that without robust pretraining, this does not hold true.
>
> 2. Yes, good catch. We have added a clear distinction between the time of neural activity and the diffusion processing. The time steps we refer to are specifically the diffusion time steps.
>
> 3. We added a detailed comparison in Appendix A
>
> We hope the reviewer found our rebuttal helpful. If there is anything we can further clarify, we are happy to do so!

---

> > ### Author Response · Authors · 2024-11-22
> >
> > Dear Reviewer,
> >
> > We kindly request your feedback on our rebuttal before the discussion period ends, which is approaching soon. Your timely input would be invaluable for a productive discussion that supports the decision-making process.
> >
> > We have made every effort to address the comments and suggestions from the initial review, and we would greatly appreciate any feedback you could provide on our revisions.
> >
> > Thank you for considering our request.

---

> > ### Comment · Reviewer_SWNx · 2024-11-25
> >
> > Thank you to the authors for the detailed responses and for incorporating additional experiments as suggested. Including the baselines, POYO and NDT, in the evaluation is helpful. However, given that the primary focus of the paper is on forecasting, I find it unclear why the new updates are compared in filtering rather than forecasting. As highlighted by other reviewers, NDT is also trained with causal masking, which suggests it should be capable of performing forecasting. I would like to see a clearer demonstration of how causal masking is insufficient and how diffusion represents a key contribution. Otherwise, I remain concerned that the current approach may appear incremental, extending from one-step to multi-step forecasting, with the role and significance of diffusion not fully evaluated.

---

> > > ### Author Response · Authors · 2024-11-25
> > >
> > > We thank the reviewer for the feedback and for raising these important points.
> > > We would like to address the concerns regarding the evaluation focus and the role of causal masking in NDT2 and our model:
> > >
> > > 1. **Comparison with NDT2:** NDT2 is not trained with a causal mask by default; rather, it is trained with a non-causal mask. For the sake of comparison and to address feedback from other reviewers, we conducted experiments to evaluate NDT2 with a causal mask. The results of these experiments are now included in Figure 3.B of the revised manuscript.
> > >
> > > | Model  | $R^2$ (hand velocity) 500 ms forecasting |
> > > |------------------|-----------------|
> > > | LFADS  |   0.34 ± 0.05 |
> > > | DVBF   |  0.33 ± 0.04|
> > > | XFADS   |  0.74 ± 0.04 |
> > > | NDT2  |0.30 ± 0.08  |
> > > | Ours  |  0.78 ± 0.05|
> > >
> > > The poor performance of NDT2 with a causal mask was expected, since recent work on transformer architectures has shown that learning global dependencies in temporal tokens leads to poor prediction results on multivariate time series datasets [1]. Thus, this model is not competitive in terms of forecasting and, in fact, that was not its original purpose, nor does POYO's architecture allow for forecasting, so our model is the only unified model proven to be able to so competitively with state-of-the-art neural data forecasting models.
> > >
> > > 2.  **Diffusion Training:**  To test whether the improved performance is due to our architecture or the training strategy, we trained the model without it (details in Appendix B.2). We found that the architecture itself is better and that training with diffusion improves it.
> > >
> > > | Model  | $R^2$ (hand velocity) 500 ms forecasting |
> > > |------------------|-----------------|
> > > | NDT2  |0.30 ± 0.08  |
> > > | Ours - diffusion training |  0.78 ± 0.05|
> > > | Ours - without diffusion training |  0.55 ± 0.04|
> > >
> > > In Fig.1 of the Appendix B.6 we show that training without diffusion required substantially more training iterations and consistently yielded lower prediction accuracy. These findings align with previous studies emphasizing the importance of intermediate noise steps in diffusion training for enhancing model prediction quality [2].
> > >
> > > 3.  **Contribution:** We would like to emphasize that our primary goal is forecasting, and previous unified models either cannot perform forecasting or their performance is not competitive with single-session neural data forecasting. To address this limitation, we have developed a distinct architecture and training strategy. Our model is specifically designed to handle both long-term and short-term dependencies for accurate forecasting. Additionally, we improved the performance and training efficiency through the use of diffusion.
> > >
> > > Given these updates, we hope the revisions address the concerns raised and provide a clearer demonstration of the contributions and significance of our work. We would greatly appreciate it if you could consider revisiting your evaluation in light of these improvements.
> > >
> > > [1] Liu, Y., Hu, T., Zhang, H., Wu, H., Wang, S., Ma, L., & Long, M. (2023). itransformer: Inverted transformers are effective for time series forecasting. arXiv preprint arXiv:2310.06625.
> > >
> > > [2] Nichol, A. Q., & Dhariwal, P. (2021, July). Improved denoising diffusion probabilistic models. In International conference on machine learning (pp. 8162-8171). PMLR.

---

> > > > ### Comment · Reviewer_SWNx · 2024-11-25
> > > >
> > > > I appreciate the authors' efforts to incorporate my suggestions and improve their work within such a short timeframe. The addition of the new experiment comparing the adapted NDT with causal masking, as well as the ablation study on diffusion in the proposed approach, greatly strengthens the paper. As a result, I would like to raise my score to 6.

---

### Official Review · Reviewer_98e5 · 2024-11-04

**Soundness:** 2
**Presentation:** 2
**Contribution:** 2
**Rating:** 3
**Confidence:** 4

**Summary:**

The paper presented a conditional diffusion model that is trained on behavioral and neural activity. The presented method can be used to forecast neural activities across animals and sessions. The paper also introduced a new benchmark consists of multiple existing datasets of neural activities.

**Strengths:**

Combining datasets to build novel benchmarks for cross-animal and cross-session pre-training is an important topic for the computational neuroscience community.

**Weaknesses:**

1. The presented method lack insights.
- Why would the authors want to forecast neural activities? How long can one forecast neural activities accurately is quite of a neuroscience topic under debate. How do the authors make sure that the problem is not ill-defined (not just model overfitting from here and there) from the beginning?
- Why would you want to embed session information at all, at test time, how could one assume they will definitely have session information? Moreover, isn't that information leakage to the prediction?

2. The presented method is confusing and seems wrong given lousy mathematical notations.
- D= {xi}, please define I here.
- Please write vectors and matrix in their corresponding math form, do not just use lowercase italicize unbold text for every notations.
- Is it a hard requirement in your framework that behavior covariates need to have a length of t? If so, clearly write this out and state it is your assumption.
- Why x is suddenly bolded in line 202? Is this the same x as previously defined? Actually, if your x is the set of neural activity and behaviors of the i-th trial, how would you plug it inside equation 1?
- Are the authors aware that diffusion is a way to train models, and a transformer is just a model architecture, and they are independent of each other? Why the authors start to introduce cross attention at line 224 without background information? What is the methodological novelty here?

3. Experimental results are confusing
- Why the authors would like to introduce a new dataset for their model evaluation? How large is the new dataset comparing to existing benchmarks? What is the advantage or uniqueness of it, and can the authors prove the advantage using experimental results? If none of the above is considered, I highly suggest the authors to split this submission to papers, where one of them focuses on presenting the new benchmark, and the other one demonstrating the method. Combining both of them makes it hard to evaluate the presented approach.
- The authors somehow decide not to benchmark with many of the methods discussed in related works. For single session results, the authors should at least include NDT [1], or their later variants such as STNDT [2] or EIT [3], where the latter two can also be extended to be transferred to new sessions.
- To be honest, it does not seem the model actually performs well. Session IDs are actually harmful for performance in single session validations, and the improvements in multi-session validations are marginal (within standard deviations).
- A common criticism in such “foundation models” on small datasets works, is that, if your single-session model is properly trained at all [4]. For example, have the authors fixed the number of training steps instead of epochs for fair evaluations? Have the authors introduced sufficient amount of augmentations? Have the authors considered using synthetic datasets for pre-training, or just initialize the model differently?


[1] Ye, J., & Pandarinath, C. (2021). Representation learning for neural population activity with Neural Data Transformers. arXiv preprint arXiv:2108.01210.

[2] Le, T., & Shlizerman, E. (2022). Stndt: Modeling neural population activity with spatiotemporal transformers. Advances in Neural Information Processing Systems, 35, 17926-17939.

[3] Liu, R., Azabou, M., Dabagia, M., Xiao, J., & Dyer, E. (2022). Seeing the forest and the tree: Building representations of both individual and collective dynamics with transformers. Advances in neural information processing systems, 35, 2377-2391.

[4] Amos, I., Berant, J., & Gupta, A. (2023). Never train from scratch: Fair comparison of long-sequence models requires data-driven priors. arXiv preprint arXiv:2310.02980.

Overall, it is quite challenging for me to decipher what is the key idea/message of the manuscript. The paper also seems like an early draft to me.

**Questions:**

NA

---

> ### Author Response · Authors · 2024-11-18
>
> We would like to direct the reviewer’s attention to the overall comment section, where we provided a detailed summary of the improvements made to both the main manuscript and the appendix. We have made significant revisions to address the reviewer’s concerns, including a more comprehensive discussion of the results, a clearer comparison with existing models (such as POYO, NDT, and EIT), and an expanded explanation of our network architecture. Additionally, we have refined the mathematical descriptions for clarity.
>
> Regarding the insights, we would like to emphasize that forecasting neural activity serves as a way to confirm that the model has learned a good approximation of the true underlying neural dynamics. While a model might excel in next-step decoding (filtering), this does not necessarily mean it has learned the correct dynamics. The use of session embeddings enables the alignment of multiple contexts by learning a shared, common representation while capturing session-specific "styles" through these embeddings. This allows our model to better generalize across different sessions.
>
>
> Finally, further motivation for the benchmark dataset can be found in the section titled "Tools for Foundation Models: Benchmark Datasets," with additional details provided in Appendix C to E. In brief, there is currently no benchmark designed specifically for multi-session models. The most similar dataset is NLB, which was designed for single-session models. However, for a fair comparison of large pre-trained models, they should be trained on the same dataset. While we acknowledge that NLB can always be used for performance comparison, using different datasets for model pretraining would not lead to a fair evaluation. Therefore, we argue that a large-scale and accessible dataset is essential for advancing the field and enabling proper comparisons across models.
>
> We hope the reviewer found our rebuttal helpful. If there is anything we can further clarify or do to improve the reviewer's overall assessment of our work, please let us know.

---

### Official Review · Reviewer_9rag · 2024-11-04

**Soundness:** 3
**Presentation:** 2
**Contribution:** 2
**Rating:** 5
**Confidence:** 3

**Summary:**

In this work, the authors propose a diffusion framework to model population dynamics of neurons recorded from multiple brain areas, tasks, subjects, and sessions. Unlike previous work in this area, their approach can generalize to new data (either new sessions from the same subject, or a new subject entirely) without the need for explicit alignment between train and test data. To demonstrate their approach, the authors have also curated a large neural dataset that can serve as a benchmark for developing models of neural population dynamics. This is a timely contribution given the growing interest in developing foundation models for neuroscience.

**Strengths:**

- The approach accounts simultaneously for neural and behavioral data with a shared latent representation
- The framework is applied to a diverse set of recordings, spanning brain areas, sessions, individuals, and tasks
- The paper is properly motivated and it is clear what problem the authors are trying to address

**Weaknesses:**

- While I can appreciate the performance improvement (as captured by the R-squared) over existing methods for modeling neural population dynamics, an analysis of what exactly the model is learning is noticeably absent. For example, the model uses a shared representation for both neural and behavioral variables; what is the relative importance of these two in forecasting neural activity? Can you perform an ablation study and compare performance with and without the behavioral data? Can you visualize the learned latent representations? Having this analysis would really improve the quality of the paper.

**Questions:**

- It seems that the datasets that constitute the benchmark utilize very structured tasks (there is a target onset, there is a go cue, etc). How well would this method work for less structured, more naturalistic tasks (e.g. freely moving animal)? Perhaps you already have data on this, if not, please feel free to speculate/discuss.

---

> ### Author Response · Authors · 2024-11-18
>
> - Weaknesses:
>
> We thank the reviewer for the suggestion to explore what the model is learning. In response, we conducted an occlusion sensitivity study, which allowed us to identify the behavior-relevant neurons and determine the specific time windows during which neuronal activity encodes movement-related information.
>
> Additionally, we carried out the recommended ablation study to compare the model's performance with and without behavioral data (Fig. 6, first plot). The results show that the joint model outperforms the two-step model in behavior prediction while preserving spiking activity accuracy. This demonstrates that the joint model effectively integrates spiking and behavior variables, optimizing both simultaneously. Unlike the two-step model, which separates these tasks, the joint model better captures their interactions, leading to improved performance without sacrificing spike prediction accuracy.
>
> We have added visualizations of the session embeddings for the multi-animal case (Figure 4.B). These embeddings cluster according to behavioral strategy, even though the model was not explicitly trained to do so. Two distinct strategies were identified: (1) fast reaction times with higher rates of failed trials, and (2) slower reaction times with higher task success. Notably, the shared session ID is positioned between the two clusters, indicating that the model successfully learned a shared representation.
>
> - Questions:
>
> That's an interesting question! Currently, we don't have data on naturalistic tasks, with the least structured task being the RTT, where the model performed well (Fig. 5.D). In the context of naturalistic tasks, we speculate that the current session embedding approach might not be directly applicable. However, we believe that we could adapt this approach by conditioning it to distinguish between different animal movements (e.g., left/right).
>
> Our hypothesis is that an initial step would involve training an embedding specifically suited for freely moving animals, which could help the model adapt to such scenarios. For example, using CLIP-style embeddings, where we provide descriptions of the animals' movements, and applying a contrastive learning approach to correlate these descriptions with neural activity and behavior. This could potentially align with techniques like CEBRA to capture the relationships in naturalistic settings.
>
> We hope the reviewer found our rebuttal helpful. If there is anything we can further clarify or do to improve the reviewer's overall assessment of our work, please let us know.

---

> ### Author Response · Authors · 2024-11-25
> **New experiments and comparisons**
>
> We would like to highlight that we conducted additional experiments to emphasize the differences between our proposed approach and previous unified models.
>
> We ran new experiments to address the previous concerns:
>
>   **Diffusion Training:**  To test whether the improved performance is due to our architecture or the training strategy, we trained the model without it (details in Appendix B.2). Our results show that the architecture itself is inherently superior, and that training with diffusion further enhances its performance.
>
> | Model  | $R^2$ (hand velocity) 500 ms forecasting |
> |------------------|-----------------|
> | LFADS  |   0.34 ± 0.05 |
> | DVBF   |  0.33 ± 0.04|
> | XFADS   |  0.74 ± 0.04 |
> | NDT2  |0.30 ± 0.08  |
> | Ours - diffusion training |  0.78 ± 0.05|
> | Ours - without diffusion training |  0.55 ± 0.04|
>
> The poor performance of NDT2 with a causal mask was expected, since recent work on transformer architectures has shown that learning global dependencies in temporal tokens leads to poor prediction results on multivariate time series datasets [1]. Thus, this model is not competitive in terms of forecasting and, in fact, that was not its original purpose, nor does POYO's architecture allow for forecasting, so our model is the only unified model proven to be able to so competitively with state-of-the-art neural data forecasting models.
>
> In Fig.1 of the Appendix B.6 we show that training without diffusion required substantially more training iterations and consistently yielded lower prediction accuracy. These findings align with previous studies emphasizing the importance of intermediate noise steps in diffusion training for enhancing model prediction quality [2].
>
>  **Contribution:** We would like to emphasize that our primary goal is forecasting, and previous unified models either cannot perform forecasting or their performance is not competitive with single-session neural data forecasting. To address this limitation, we have developed a distinct architecture and training strategy. Our model is specifically designed to handle both long-term and short-term dependencies for accurate forecasting. Additionally, we improved the performance and training efficiency through the use of diffusion.
>
> Given these updates, we hope the revisions address the concerns raised and provide a clearer demonstration of the contributions and significance of our work. We would greatly appreciate it if you could consider revisiting your evaluation in light of these improvements.
>
> [1] Liu, Y., Hu, T., Zhang, H., Wu, H., Wang, S., Ma, L., & Long, M. (2023). itransformer: Inverted transformers are effective for time series forecasting. arXiv preprint arXiv:2310.06625.
>
> [2] Nichol, A. Q., & Dhariwal, P. (2021, July). Improved denoising diffusion probabilistic models. In International conference on machine learning (pp. 8162-8171). PMLR.

---

### Official Review · Reviewer_Dja5 · 2024-11-04

**Soundness:** 2
**Presentation:** 1
**Contribution:** 2
**Rating:** 3
**Confidence:** 4

**Summary:**

This paper introduces the idea of using causal diffusion modelling to jointly model neural and behavioral data across large number of ephys recordings. This approach is applied for two tasks---neural activity forcasting and behavior decoding. Additionally, the paper also collates a set of public datasets to form a large-dataset to be used as a benchmark for pretraining neural population decoding models.

**Strengths:**

- Use of diffusion as an SSL framework for pretraining of large neuroscience models is a novel idea.
- Collation of public ephys recording data to create a large scale benchmark is a good contribution to the field.

**Weaknesses:**

In its current state, this paper should be rejected because:
1. The paper fails to demonstrate the effectiveness of using diffusion as a pretraining strategy in comparison to other modern pretraining strategies such as NDT2 [1] and POYO [2]
2. The paper is unpolished and lacks many details important for replication of the presented method.

**Questions:**

**Arguments on experiments conducted and results:**
1. Since the majority of results in this work are on behavior decoding, I will assume that is an important metric that this paper uses to compare the presented method with baselines. Although it is positive to see that training on more data improves decoding performance, this positive effect of data scale for behavior decoding has been established by NDT2 and POYO. I believe, in current environment, it's important to see these performance numbers in relation to other modern large-scale training methods such as NDT2 and POYO. Without such comparisons, an important scientific remains unanswered----does diffusion have any benefits to offer over other large-scale pretraining techniques like supervised pretraining or mask-modelling for behavior decoding? If diffusion is not comparable or better in comparison, then the benefits of using it for the purpose of behavior decoding have not been demonstrated.
2. Even when considering neural forecasting as a task, the lack of results in relation to NDT2 is a major drawback. Since NDT2 is trained using a causal masked-modelling framework, it is also capable for performing forecasting. And hence, to make and argument for diffusion, a comparison to masked-modelling like NDT2 should also be presented.

**Additional comment on comparison to NDT2/POYO:**
I would understand if training NDT2 and/or POYO on your dataset might by prohibitive due to the scale and compute intensiveness of those methods. In this case, I would suggest performing the following analysis: Do multi-session training on all sessions except the ones included in the NLB Maze and NLB RTT benchmarks [3] (the sessions from these two benchmarks seem to already be part of the dataset considered in this paper). Then, do transfer on these two benchmarks to show your method compares to the baselines (I believe both NDT2 and POYO mention performance on these datasets).



**On missing details about method:**
1. How is a sample $x_i$ passed into the U-Net? Since the number of neurons change from one sample to another in the multi-session training situation, it is implied that the dimensionality of $x_i$ changes as well. It is not clear to me how a fixed-sized CNN is used to process all the samples simultaneously.
2. Figure 1 shows multiple session-embedding tokens going in the Key/Value port of the cross-attention for _each_ sample. Since there are multiple tokens, how are they differentiated from one another? Do these tokens have any position embedding added? If so, of which kind?
3. It is unclear as to how the equation on line 265 is used to enable/disable session guidance. The symbol $s$ or the term "_guidance scale_" is not used anywhere else in the text.
4. Since you forcast a 500ms window, how do you perform behavior decoding for 1s trials? (most datasets under consideration in here have trial-durations of ~1s)

As a general comment, I would encourage the authors to make the model's description in the paper as self-contained and complete as possible. Especially since it states the model as being a core contribution.

**Some smaller concerns/questions:**
1. Why not include full-finetuning as a transfer strategy, in addition to zero-shot and session-identification?
2. I regards to table 2, I am curious about your explanation as to why performing Session-ID does worse than zero-shot in the case of single-session training?
3. Figure 4C: What's the difference between "All Sessions" and "Full dataset"?
4. Figure 4D: both the red and blue curves look "too smooth" for being raw $R^2$ curves. I am curious if this is a result of performing some post-processing on the raw curves, such as applying smoothing? If so, I would suggest not doing much smoothing as there is value in being able to look at the raw behavior of accuracy changing over training steps.
5. Regarding figure 4C - this figure about data scaling, and hence the x-axis should be changed to number of recording-hours or another relevant data-scale metric and not model parameters.

[1] Neural Data Transformer 2: Multi-context Pretraining for Neural Spiking Activity. Joel Ye et. al. NeurIPS 2023

[2] A Unified, Scalable Framework for Neural Population Decoding. Mehdi Azabou et. al., NeurIPS 2023

[3] https://neurallatents.github.io/

---

> ### Author Response · Authors · 2024-11-18
>
> Weaknesses:
>
> 1. We thank the reviewer for the suggestion to run more baseline experiments. However, the comparisons with NDT2 and POYO are not directly feasible because our proposed model is a unified framework capable of forecasting, whereas NDT2 and POYO were specifically designed for filtering tasks—i.e., predicting only the next time step based on a history of neural activity. This distinction was the main reason for not including these comparisons, and we apologize if the original manuscript was unclear on this point. To address this, we have now provided a clearer explanation in the methods section (C.5) and in Figure 1 of the revised manuscript.
> Based on the reviewer’s feedback, we conducted a new experiment where we adapted our model to a filtering setting, predicting only the next time step. Unfortunately, NDT2 does not provide results for NLB datasets. In this new experiment, we compared our model against several other methods, including the Wiener filter, GRU, MLP, AutoLFADS, NDT, NDT-Sup, EIT, XFADS, and POYO on the NLB-maze dataset (Figure 1.C in the updated version). Additionally, to compare our model with other modern pretraining strategies, we ran experiments on the RTT dataset, comparing the performance of training from scratch versus training with the full dataset for NDT2 and POYO (Figure 5.D in the new version).
> While POYO remains the best-performing model in this comparison, our model is highly competitive, outperforming NDT2. However, it is important to note that filtering is not the primary focus of our model. A detailed, though not exhaustive, summary of the differences between our approach and others has been added to Appendix A. Finally, we would like to emphasize that POYO is not capable of forecasting, which represents an unexplored area in the NDT2 framework.
>
> 2. We agree with the reviewer that the network architecture is an important detail missing from our paper.  To address this concern, we have included a full description of the model in Section C.2 and additional details in Appendix B. We hope that the updated manuscript fully clarifies any remaining questions related to the model. Nevertheless, we will address the missing details of the method here:
>  - Each neural population in the model has a different number of neurons, which can pose a challenge when working with multiple sessions. To address this, we standardize the input size by aligning all populations to the size of the largest population, padding the smaller populations as needed. It is important to note that, similar to transformers, the padded portions of the input are ignored during training by the loss function, ensuring that the padding does not influence the model's learning process (Appendix B.4).
> - Each sample (trial) is associated with a session ID, which links it to a specific session. The session IDs are represented as a learnable lookup table, where each ID (scalar) corresponds to a high-dimensional vector. This vector is then used to guide the generation process through cross-attention, in a manner similar to how CLIP embeddings guide the text-to-image generation in diffusion models.
> - We agree with the reviewer that the distinction between conditional and unconditional models was not well defined in the original manuscript. The model is considered unconditional when using the shared session ID. The equation is only applied during sampling in the conditional case, where we generate a sequence conditioned on a specific session ID. The conditioning is achieved by assigning 90% of the model’s focus to the chosen session ID and 10% to the shared one. This guidance scale, which controls how much attention the model gives to the session ID, is a hyperparameter during the sampling phase.
> - The model autoregressively predicts an arbitrarily long sequence by using the previously generated sequence as input for predicting the next steps.

---

> > ### Author Response · Authors · 2024-11-18
> >
> > Smaller concerns/questions:
> > 1. We appreciate the reviewer’s suggestion to include full fine-tuning as a transfer strategy. In response, we ran additional experiments to compare the three transfer strategies, which are now presented in Table 1 of the revised manuscript.
> >
> > 2. Yes, this is a good question, which provided an opportunity to clarify the results. In response, we have added our explanation and hypothesis regarding the transfer strategies: 'Our transfer strategies aim to learn shared features while also capturing session-specific styles through embeddings. However, training on a single small session is insufficient for building a noise-robust model. Few-shot examples from a different animal fail to separate shared and session-specific features, which pushes the model further away from the data manifold and amplifies noise in the learned representation. As noted by Vermani et al. (2024), the current training session itself is highly informative about the testing sessions, which we hypothesize explains why zero-shot learning performs better in this scenario.
> > With sufficient data, the model can learn robust shared features, while few-shot examples help capture session-specific styles, ultimately improving performance. This emphasizes the importance of having adequate data for building robust representations. While Azabou et al. (2023) demonstrated that large-scale pretraining allows for effective session identification adaptation, our results suggest that without robust pretraining, this does not hold true.'
> >
> > 3.  To clarify, when we refer to "all sessions," we mean "multi-session," which refers to all sessions of an animal performing a particular task.
> >
> > 4. Yes, the curves represent the mean values.
> >
> > We hope the reviewer found our rebuttal helpful. If there is anything we can further clarify or do to improve the reviewer's overall assessment of our work, please let us know.

---

### Author Response · Authors · 2024-11-18
**Thank you for your time and constructive comments.**

We thank the reviewers for their thoughtful and detailed feedback on our manuscript. We are excited to know that the reviewers considered our work ‘ is developed with great accessibility including introducing a benchmark dataset and APIs to benefit the broader community.’ (SWNx), ‘Use of diffusion as an SSL framework for pretraining of large neuroscience models is a novel idea.’ (Dja5) and ‘the results presented are encouraging’ (hXMN). And that  ‘The paper is properly motivated and it is clear what problem the authors are trying to address’ (9rag).

In the rebuttal, we did our best to address any concerns. To this end, we carried out several new experiments and updated the main manuscript and the Appendix, as described below:
- Filtering experiments:

We broaden the scope of our work to demonstrate that, in addition to forecasting, our model can be compared to previous unified models designed for filtering tasks, such as POYO and NDT2. These models operate by predicting the next time step of behavioral variables given a history window of neural activity, incrementally sliding this window to produce the desired sequence of predicted behaviors. To the best of our knowledge, ours is the only unified model that has proved capable of forecasting.
We evaluated the performance of our model against the most recent state-of-the-art methods: XFADS for single-session tasks and SeqVAE for multi-session tasks. Additionally, based on feedback from multiple reviewers, we adapted our model to perform filtering to assess its effectiveness in this context. Results indicate that our framework is as competitive as NDT2 and POYO, even though filtering is not its primary focus and the design was not optimized specifically for this task.
Traditionally, diffusion models excel in full-sequence generation rather than next-token prediction, which is a strength of transformer-based architectures. Nonetheless, these new experimental results have been incorporated into the main manuscript (Figures 3.C and 5.D), with a more detailed comparison between our approach and other foundational models provided in Appendix A.

- More details on Model architecture:

We have added a new section (C.2) to provide a step-by-step explanation of the model architecture. Additionally, we describe the implementation of the conditional and unconditional components of the model in section C.4, and include all the hyperparameters used in Appendix B.

- Interpretation:

We conducted an occlusion sensitivity study to identify behavior-relevant neurons and determine the specific time windows during which neuronal activity encodes movement-related information (Results: Fig. 3.D; additional details in Appendix B.6).
Additionally, we included visualizations of session embeddings for the multi-animal case (Figure 4.B). These embeddings clustered according to behavioral strategy, despite no explicit training for this outcome. Two distinct strategies were identified: (1) fast reaction times associated with higher rates of failed trials, and (2) slower reaction times with greater task success. Notably, the shared session ID is positioned between the two clusters, indicating that the model effectively learned a shared representation.

- Joint vs non joint train:

Panel A of Fig. 6 (new version) compares our joint model with the two-step model, where behavior is predicted using ridge regression based on next-step spiking activity. The results show that the joint model outperforms the two-step model in behavior prediction while preserving spiking activity accuracy. This demonstrates that the joint model effectively integrates spiking and behavior variables, optimizing both simultaneously. Unlike the two-step model, which separates these tasks, the joint model better captures their interactions, leading to improved performance without sacrificing spike prediction accuracy.

- Ablation Studies:

We conducted several ablation studies to assess the impact of training with both spiking activity and behavioral data, as well as the effects of data preprocessing and forecasting capabilities (Fig. 6). These studies suggest that the model is robust to data preprocessing, demonstrates strong forecasting capabilities, and further confirm our previous results.

---

> ### Author Response · Authors · 2024-11-25
> **New experiments and comparisons**
>
> **Contribution:** We would like to emphasize that our primary goal is forecasting, and previous unified models either cannot perform forecasting or their performance is not competitive with single-session neural data forecasting. To address this limitation, we have developed a distinct architecture and training strategy. Our model is specifically designed to handle both long-term and short-term dependencies for accurate forecasting. Additionally, we improved the performance and training efficiency through the use of diffusion.
>
> **Comparison with NDT2 with causal mask:** NDT2 is not trained with a causal mask by default; rather, it is trained with a non-causal mask. For the sake of comparison and to address feedback from the reviewers, we conducted experiments to evaluate NDT2 with a causal mask. The results of these experiments are now included in Figure 3.B of the revised manuscript.
>
> | Model  | $R^2$ (hand velocity) 500 ms forecasting |
> |------------------|-----------------|
> | LFADS  |   0.34 ± 0.05 |
> | DVBF   |  0.33 ± 0.04|
> | XFADS   |  0.74 ± 0.04 |
> | NDT2  |0.30 ± 0.08  |
> | Ours  |  0.78 ± 0.05|
>
> The poor performance of NDT2 with a causal mask was expected, since recent work on transformer architectures has shown that learning global dependencies in temporal tokens leads to poor prediction results on multivariate time series datasets [1]. Thus, this model is not competitive in terms of forecasting and, in fact, that was not its original purpose, nor does POYO's architecture allow for forecasting, so our model is the only unified model proven to be able to so competitively with state-of-the-art neural data forecasting models.
>
>  **Diffusion Training:**  To test whether the improved performance is due to our architecture or the training strategy, we trained the model without it (details in Appendix B.2). We found that the architecture itself is better and that training with diffusion improves it.
>
> | Model  | $R^2$ (hand velocity) 500 ms forecasting |
> |------------------|-----------------|
> | NDT2  |0.30 ± 0.08  |
> | Ours - diffusion training |  0.78 ± 0.05|
> | Ours - without diffusion training |  0.55 ± 0.04|
>
> In Fig.1 of the Appendix B.6 we show that training without diffusion required substantially more training iterations and consistently yielded lower prediction accuracy. These findings align with previous studies emphasizing the importance of intermediate noise steps in diffusion training for enhancing model prediction quality [2].
>
>
>
> [1] Liu, Y., Hu, T., Zhang, H., Wu, H., Wang, S., Ma, L., & Long, M. (2023). itransformer: Inverted transformers are effective for time series forecasting. arXiv preprint arXiv:2310.06625.
>
> [2] Nichol, A. Q., & Dhariwal, P. (2021, July). Improved denoising diffusion probabilistic models. In International conference on machine learning (pp. 8162-8171). PMLR.

---

### Meta-Review · Area_Chair_jWDT · 2024-12-21

**Metareview:**

This paper proposes to tackle the challenge of learning a unified model to forecast neural activity and behavior across animals, sessions, and tasks. A conditional diffusion model is trained to perform this task (conditioned on session identifiers based on prior work), trained with a causal mask. The authors perform experiments on a combined ephys dataset (with 6 subdatasets of 19 monkeys total across 7 tasks). Experiments are performed on single-session predictions, new-session transfer, and across tasks -- showing that the proposed approach is either comparable, or better than, baselines such as POYO and NDT2 (mainly for the "filtering" task, or single-step prediction). Reviewers appreciate the importance of the task tackled by the paper and the proposed diffusion model approach. However, concerns were raised on comparison of this approach to baselines, lack of analysis on what the model is learning, and unclear aspects of the approach. There is also some confusion from reviewers and in the text on forecasting vs. filtering, which seems to have been resolved by the rebuttal for some reviewers. However, since this is a key difference between the proposed approach and baselines, it would be valuable to run previous "filtering" approaches in a sliding window to compare forecasting performance. The AC encourages the authors to further refine the text so the contribution of this work is more clear (e.g. transformers can also be used for autoregressive generation to "forecast" & generate videos - VideoPoet is an example from a different domain, but just to illustrate the concept https://arxiv.org/pdf/2312.14125). Beyond forecasting, another experiment that would significantly strengthen this work would be if the authors could demonstrate an example of an insight that is found through their framework, but is not possible through POYO or NDT2. In this current state, I think the paper is below the bar, but the ideas are interesting, and with more clear presentation and additional experiments suggested by reviewers, it could make a strong future submission.

**Additional Comments On Reviewer Discussion:**

Reviewers initially raised concerns on the lack of comparisons to baselines, suggestions on additional experiments (e.g. finetuning), dataset differences with other papers, and presentation issues. While the authors addressed some of the reviewers concerns (and some reviewers were highly engaged in the discussions), it seems that other concerns are not addressed by the rebuttal (the AC also acknowledges that some reviewers did not respond to authors, and I read the author rebuttal for those reviewers). The key issue is the relation of this paper given existing approaches in the field. While is is likely the first work to use diffusion in this setting, I think it is important to demonstrate the value of using a new framework. As is, the paper does not make a convincing case why existing approaches cannot be used for forecasting (e.g. using autogressive generation), and also, what new insights does the framework bring to neuroscience over a multi-session, multi-animal, multi-task transformer? Addressing these questions would be helpful for future versions of this work.

---

### Decision · Program_Chairs · 2025-01-22

Reject